JCB Journal of Cell Biology

# Analysis of copy number alterations reveals the lncRNA ALAL-1 as a regulator of lung cancer immune evasion

Alejandro Athie[1,2]*, Francesco P. Marchese[1,2]*, Jovanna González[1,2], Teresa Lozano[2,3], Ivan Raimondi[1,2], Prasanna Kumar Juvvuna[4], Amaya Abad[1,2], Oskar Marin-Bejar[1,2], Jacques Serizay[1,2], Dannys Martínez[1,2], Daniel Ajona[2,5,6], Maria Jose Pajares[2,5,6], Juan Sandoval[7,8], Luis M. Montuenga[2,5,6], Chandrasekhar Kanduri[4], Juan J. Lasarte[2,3], and Maite Huarte[1,2]

**Cancer is characterized by genomic instability leading to deletion or amplification of oncogenes or tumor suppressors. However, most of the altered regions are devoid of known cancer drivers. Here, we identify lncRNAs frequently lost or amplified in cancer. Among them, we found *amplified lncRNA associated with lung cancer-1* (*ALAL-1*) as frequently amplified in lung adenocarcinomas. *ALAL-1* is also overexpressed in additional tumor types, such as lung squamous carcinoma. The RNA product of *ALAL-1* is able to promote the proliferation and tumorigenicity of lung cancer cells. ALAL-1 is a TNFα– and NF-κB–induced cytoplasmic lncRNA that specifically interacts with SART3, regulating the subcellular localization of the protein deubiquitinase USP4 and, in turn, its function in the cell. Interestingly, *ALAL-1* expression inversely correlates with the immune infiltration of lung squamous tumors, while tumors with ALAL-1 amplification show lower infiltration of several types of immune cells. We have thus unveiled a pro-oncogenic lncRNA that mediates cancer immune evasion, pointing to a new target for immune potentiation.**

## Introduction

Most cancers arise and later progress due to the complex interaction of somatic and germline mutations with various environmental factors. Many of these mutations lie within regions of the genome devoid of protein-coding genes (Maurano et al., 2012), which may contain different types of genes that exert their functions as RNA molecules, the noncoding RNAs. Most noncoding RNAs are longer than 200 nucleotides and are therefore classified as long noncoding RNAs (lncRNAs). The number of lncRNAs encoded by human cells is large, cataloged in a range that spans from 16,000 genes encoding close to 28,000 transcripts—according to the estimation of GENCODE (Derrien et al., 2012; Djebali et al., 2012)—to >60,000 lncRNA transcripts identified across multiple tumor types (Iyer et al., 2015). A number of studies have shown that many lncRNAs are functional molecules that can regulate different aspects of cell biology through multiple mechanisms (Engreitz et al., 2016; Rinn and Chang, 2012). In agreement with this, we and others

have observed that alterations in lncRNAs are inherent to cancer and can impact several hallmarks of the disease (reviewed in Gutschner and Diederichs, 2012; Huarte, 2015; Pasut et al., 2016; and Schmitt and Chang, 2016). However, despite the high number of lncRNAs encoded by the human genome, our understanding of their contribution to the disease remains poor. Moreover, while identification of relevant cancer-driver lncRNAs is necessary to better understand tumor progression, it represents a major challenge due to different reasons. For instance, a high percentage of the thousands of uncharacterized lncRNAs present altered expression in cancer (Iyer et al., 2015), but most of them are possibly passenger alterations. Furthermore, the high heterogeneity of cancer and the tissue specificity of lncRNAs complicate the identification of lncRNA alterations relevant to a specific cancer type.

Here, we analyzed >7,000 tumors of 25 different types of cancer in order to detect the genomic copy number alterations in

[1]Department of Gene Therapy and Regulation of Gene Expression, Center for Applied Medical Research, University of Navarra, Pamplona, Spain; [2]Institute of Health Research of Navarra, Pamplona, Spain; [3]Department of Immunology and Immunotherapy, Center for Applied Medical Research, University of Navarra, Pamplona, Spain; [4]Department of Medical Biochemistry and Cell Biology, Institute of Biomedicine, Sahlgrenska Academy, University of Gothenburg, Gothenburg, Sweden; [5]Department of Solid Tumors, Center for Applied Medical Research, University of Navarra, Pamplona, Spain; [6]Department of Pathology, Anatomy and Physiology, University of Navarra and CIBERONC, Centro de Investigación Biomédica en Red de Cáncer, Madrid, Spain; [7]Biomarkers and Precision Medicine Unit, Health Research Institute La Fe, Valencia, Spain; [8]Epigenomics Core Facility, Health Research Institute La Fe, Valencia, Spain.

*A. Athie and F.P. Marchese contributed equally to this paper; Correspondence to Maite Huarte: maitehuarte@unav.es; Alejandro Athie's present address is Department of Clinical Research, Vall d'Hebron Institute of Oncology, Barcelona, Spain; Oskar Marin-Bejar's present address is Laboratory for Molecular Cancer Biology, Department of Oncology, KU Leuven, Leuven, Belgium; Jacques Serizay's present address is The Gurdon Institute and Department of Genetics, University of Cambridge, Cambridge, UK.

lncRNAs positively or negatively selected during tumor progression. Our analysis led to the identification of a number of lncRNA loci frequently amplified or deleted in different cancer types. Among them, we identified and characterized *amplified lncRNA associated with lung cancer-1* (*ALAL-1*). *ALAL-1* was found amplified in lung cancer, where it showed oncogenic features and, by regulating inflammatory mediators, promoted the immune evasion of lung cancer cells.

## Results

### Several frequent cancer-associated somatic copy number alterations (SCNAs) devoid of protein-coding genes contain lncRNAs

We reasoned that lncRNAs with an oncogenic or tumor suppressor role should be positively or negatively selected in cancer genomes. To identify lncRNAs frequently amplified or deleted in cancer, we retrieved the SCNA data available from The Cancer Genome Atlas (TCGA), comprising a total of 7,448 tumors of 25 different tumor types (Fig. 1 A). To detect the potentially relevant SCNAs, we used the GISTIC 2.0 algorithm (Mermel et al., 2011), which assigns a score to each alteration based on its amplitude (copy number changes) and frequency across all samples (G-score = Frequency × Amplitude). False discovery rate q-values for the aberrant regions were then calculated, and a threshold of 0.25 was established to select significant alterations. With this method, 1,377 SCNAs were identified (540 amplifications and 837 deletions) at three different levels: region, enlarged peaks, and focal peaks (Fig. S1, A and B; and Materials and methods). While genomic instability inherent to cancer cells can lead to large SCNAs that contain thousands of genes (regions and enlarged peaks), driver alterations usually occur in regions containing only a few genes (Zack et al., 2013). For this reason, we focused the rest of our analysis on SCNAs at the focal peak level. In total, we identified 1,026 unique copy number–altered focal regions. 916 of them were specific to a tumor type, while the rest (110) were present in several tumor types (Fig. 1 B, Fig. S1 C, and Table S1).

For a comprehensive view of all the genes, coding and noncoding, affected by the copy number alterations, the SCNAs were classified based on the annotation of the genes contained in them (GENCODE v19), as well as taking into consideration their known cancer driver features (Fig. 1, B–D). Out of the 1,026 SCNAs, 136 contained a known cancer driver (Fig. 1, C and D), as defined by the high-confidence driver list previously reported (Tamborero et al., 2013). For instance, the tumor suppressors *PTEN* and *RB1* were frequently lost, while the oncogene *MYC* was inside a frequently amplified region, confirming the validity of the SCNA analysis to pinpoint genes relevant to tumor progression. On the other hand, the 890 remaining SCNAs did not contain any known cancer driver gene. The classification of these SCNAs based on the biotypes of the included genes showed that 50 of them contained only lncRNAs (Fig. 1 D), suggesting that these lncRNAs are frequently lost or amplified independently of protein-coding genes and could act as cancer drivers. In addition, 97 SCNAs did not contain any annotated gene and were therefore classified as "gene deserts."

### lncRNAs within frequent SCNAs have functional features

To increase our insight into the functional characteristics of the lncRNAs within frequent SCNAs, we analyzed their regulation by relevant transcription factors. To that end, we retrieved the transcription start site (TSS) of the copy number–altered lncRNAs and arbitrarily defined their promoter region 1 kb upstream and 1 kb downstream of the TSS. The resulting genomic coordinates were then intersected with the binding sites of 161 transcription factors obtained from chromatin immunoprecipitation (ChIP) experiments reported by ENCODE. The promoters of deleted lncRNAs showed enrichments in binding for POU5F1/OCT4 and NANOG1, indicating regulation by pluripotency-related transcription factors (Fig. S1 D). Interestingly, amplified lncRNAs were significantly enriched for oncogenic transcription factors such as MYC, MAX, and JUND (Fig. 1 E), pointing toward their specific regulation by oncogenic signals, in line with their amplified status.

We reasoned that if the genes contained within SCNAs have an impact in cancer progression, they should present a change in expression consistent with the sense of the copy number alteration (amplification or deletion). We therefore cross-compared our SCNA results with RNA expression profiling analysis in cancer. For this, we used the comprehensive annotation of cancer-associated lncRNAs (Iyer et al., 2015), identifying a total of 20 putative functional alterations (14 amplifications and 6 deletions) in which the expression levels of the contained lncRNAs agreed with the SCNA type when tumor and normal tissue were compared (i.e., higher expression levels for amplified genes and lower levels for deleted genes in tumor; Fig. 1 F). Among the regions shortlisted using our approach, we found some previously characterized cancer-related lncRNAs such as *PVT1*, localized downstream of the *MYC* locus (Tseng et al., 2014), or *CCAT1/CARLo-5* (Kim et al., 2014), located in an amplified region upstream of the *MYC* locus. However, most of the identified lncRNAs remain uncharacterized, and so far no functional role has been assigned to them.

### *ALAL-1* is a potential driver of non–small cell lung cancer (NSCLC)

Among the uncharacterized lncRNAs identified by our SCNA analysis, we focused on those found in lung cancer. The inspection of these SCNAs (CNA_202, CNA_623, and CNA_793; Fig. 1, F and G) showed that CNA_202 contains *CARLo-1/CASC8*, previously related to several types of tumors (Xiang et al., 2014; Kim et al., 2014; Knipe et al., 2014; Hu et al., 2016), while CNA_793 has also been linked to lncRNAs since it maps to the frequently deleted Prader-Willi/Angelman region containing seven noncoding RNAs (PWRN2, RP11-580I1.1, RP11-580I1.2, RP11-350A1.2, RP11-107D24.2, PWRN3, and PWRN1; Buiting et al., 2007; Stelzer et al., 2014). On the other hand, CNA_623 overlaps only the uncharacterized gene RP11-231D20.2, which is the reason we focused our attention on this amplified region (Fig. 2 A).

Analysis of the TCGA lung adenocarcinoma (LUAD) cohort showed that 43/493 (8.72%) tumors contain the alteration CNA_623. To confirm this frequency, we analyzed independent cohorts of LUAD patients—at the Center for Applied

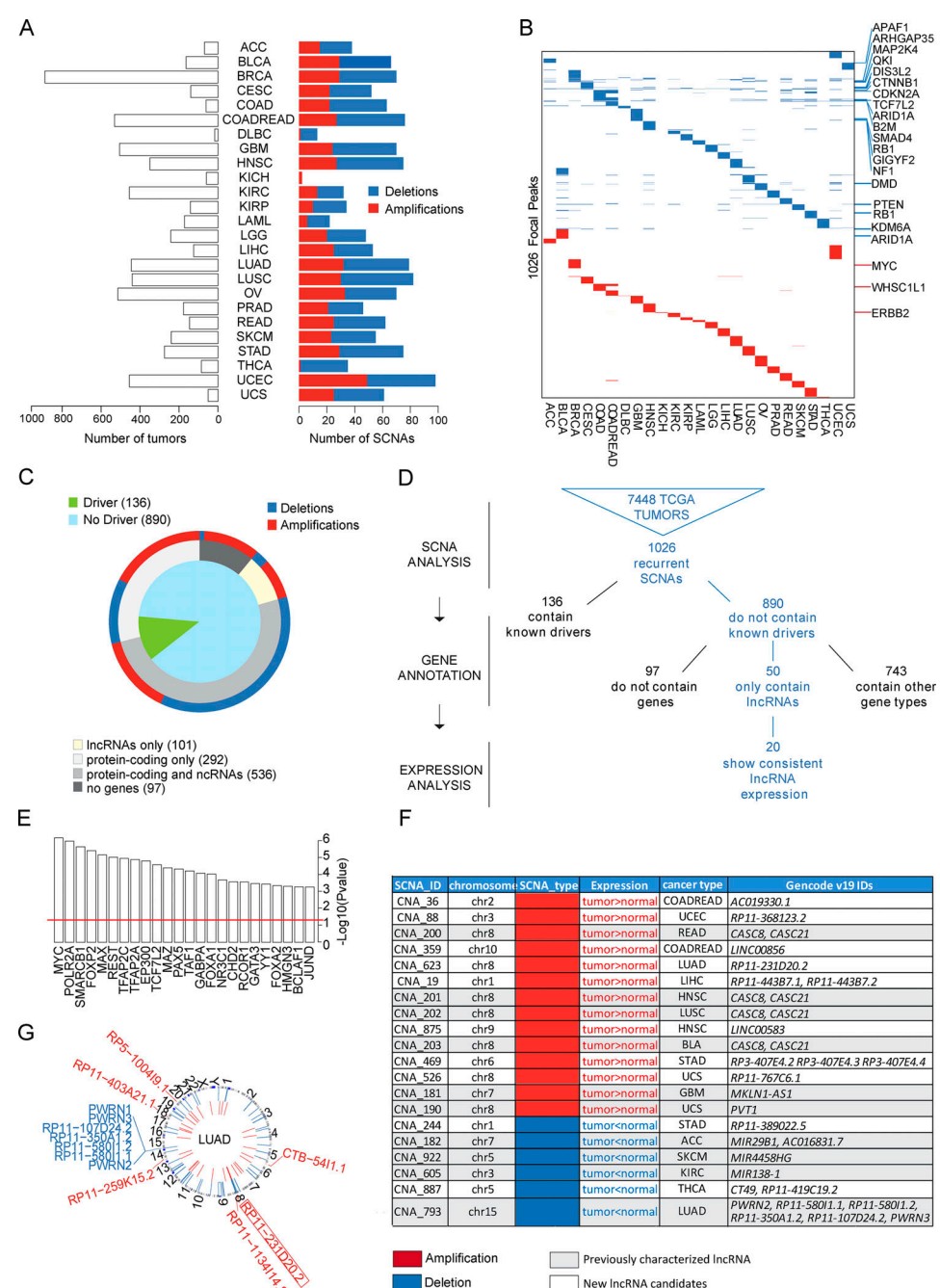

Figure 1. **Overview of cancer-associated SCNAs that contain lncRNAs. (A)** Distribution of the TCGA tumor samples and SCNAs detected in the 25 types of cancer. **(B)** Distribution of SCNAs in the different cancer types, indicating some of the high-confidence cancer drivers (Tamborero et al., 2013) within the alterations. **(C)** Classification of the SCNAs based on the biotypes of the genes inside them. **(D)** Pipeline for the selection of SCNAs harboring lncRNAs, integrating SCNA detection, gene annotation, and expression analysis. The starting point of the analysis is the copy number data from 7,448 tumors, in which 1,055 recurrent SCNAs were detected. Then these SCNAs were classified based on gene annotation and gene expression analysis. **(E)** Transcription factors with binding significantly enriched around the TSS of amplified lncRNAs. Statistical significance was determined by hypergeometric test; P values were −log10 transformed. **(F)** SCNAs selected by the analysis as containing putative cancer-relevant lncRNAs. **(G)** Circos plot indicating the copy number–altered lncRNAs present in lung adenocarcinomas. Indicated in bold and squared is the lncRNA RP11-231D20.2 selected for the present study. ncRNA, noncoding RNA; ACC, adrenocortical carcinoma; BLCA, bladder urothelial carcinoma; BRCA, breast invasive carcinoma; CESC, cervical squamous cell carcinoma and endocervical adenocarcinoma; COAD, colon adenocarcinoma; COADREAD, colorectal adenocarcinoma; DLBC, lymphoid neoplasm diffuse large B-cell lymphoma; GBM, glioblastoma multiforme; HNSC, head and neck squamous cell carcinoma; KICH, kidney chromophobe; KIRC, kidney renal clear cell carcinoma; KIRP, kidney renal papillary cell carcinoma; LAML, acute myeloid leukemia; LGG, brain lower grade glioma; LIHC, liver hepatocellular carcinoma; OV, ovarian serous cystadenocarcinoma; PRAD, prostate adenocarcinoma; READ, rectum adenocardinoma; SKCM, skin cutaneous melanoma; STAD, stomach adenocarcinoma; THCA, thyroid carcinoma; UCEC, uterine corpus endometrial carcinoma; UCS, uterine carcinosarcoma.

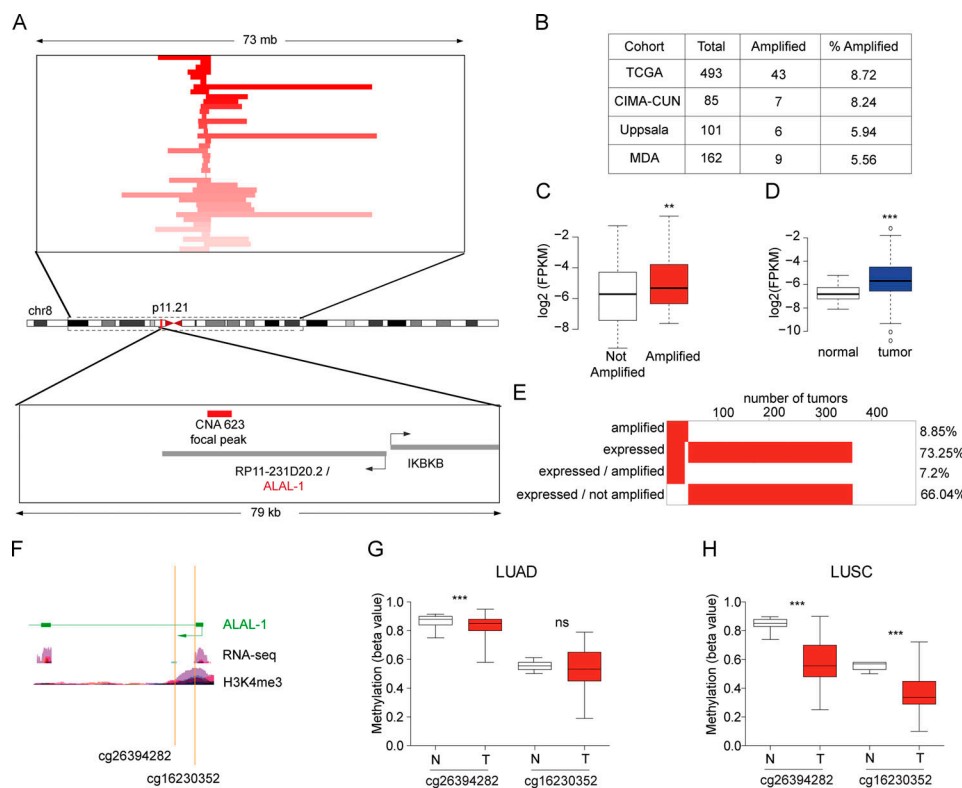

Figure 2. **ALAL-1 is a potential driver of NSCLC. (A)** Detailed view of the genomic region of the focal peak mapping to *RP11-231D20.2/ALAL-1*. Each TCGA-LUAD sample with amplification is represented with a horizontal line and ranked according to the copy number of the segment shown. The focal amplification CNA 623 defined by GISTIC 2.0 algorithm mapping to the ALAL-1 locus is indicated in red. **(B)** Number of tumors presenting amplification of *ALAL-1* in different cohorts of lung adenocarcinoma. **(C)** Expression of *ALAL-1* in TCGA-LUAD cohort comparing the samples based on the presence (*n* = 43) or absence (*n* = 322) of the amplification. **(D)** Expression of *ALAL-1* in tumor (*n* = 291) versus normal (*n* = 21) samples from TCGA-LUAD. **(C and D)** Statistical significance was determined by two-tailed unpaired *t* test with Welch's correction. **(E)** Percentage of samples with amplification and expression (FPKM > 0) of ALAL-1. **(F)** Schematic representation of the 5′ of the *ALAL-1* locus, indicating the two methylated CpGs (cg26394282 and cg16230352) and the RNA-seq and H3K4me3 ChIP-seq signals. **(G and H)** Methylation level reported using the β value for the two *ALAL-1*–associated CpGs in normal (N) and tumor (T) samples from TCGA-LUAD (G) and TCGA-LUSC cohorts (H). Bottom and top of the box are the 25th and 75th percentile (the lower Q1 and upper quartiles, respectively Q3), and the band near the middle of the box corresponds to the median. The lower whisker extends Q1 − 1.5 * interquartile range (IQR) and the upper one Q3 + 1.5 IQR. Statistical significance was determined by two-tailed unpaired *t* test with Welch's correction. Statistical significance is represented as **, P ≤ 0.01; ***, P ≤ 0.001. ns, not significant; MDA, The University of Texas MD Anderson Cancer Center. CIMA-CUN, Center for Applied Medical Research-Navarra University Clinic; FPKM, fragments per kilobase of exon model per million reads mapped; ns, not significant.

Medical Research–University of Navarra Clinic (Pamplona, Spain; Aramburu et al., 2015), Uppsala University (Uppsala, Sweden; Micke et al., 2011), and The University of Texas MD Anderson Cancer Center (Houston, TX; Aramburu et al., 2015)—where the amplification was identified in 8.24%, 5.94%, and 5.56% of the tumors, respectively (Fig. 2 B). CNA_623 uniquely maps to RP11-231D20.2, hereafter referred to as *ALAL-1*. *ALAL-1* is located in the short arm of chromosome 8 (chr8:42,091,193-42,128,429) as a divergent antisense transcript of the *IKBKB* gene. GENCODE v19 annotation shows six different transcripts associated with an ALAL-1 locus (Fig. S1, E and F). To assess which of the transcriptional forms is predominantly expressed in lung tumor cells, a TCGA-LUAD sample with a mean expression of *ALAL-1* was selected, and the RNA sequencing (RNA-seq) reads were mapped to the locus (Fig. S1 G). The RNA-seq supported the predominant expression of the 415 nt–long transcript ENST00000521802, which has three exons (Fig. S1 G). This observation was confirmed by quantitative RT-PCR (qRT-PCR) using different sets of primers (Fig. S1 H).

Based on these analyses, we focused on this transcript form of *ALAL-1*.

To evaluate the possible role of *ALAL-1* in lung cancer, we tested whether *ALAL-1* was indeed overexpressed in tumors where the amplification was present. To do this, we divided the tumor samples into two groups, with and without amplification. As expected, a significantly higher expression of *ALAL-1* was observed in the amplified group (Fig. 2 C). Moreover, the expression analysis of all the TCGA-LUAD samples (comparing normal vs. tumor) also showed a significant difference on *ALAL-1* expression levels (Fig. 2 D). In fact, around 66% of the tumor samples with higher expression of *ALAL-1* lacked the amplification of the *ALAL-1* locus (Fig. 2 E). Moreover, *ALAL-1* was also overexpressed in additional cohorts and tumor types, such as lung squamous carcinoma (LUSC) and head and neck squamous carcinoma (Fig. S2, I–L). Interestingly, the most significant difference in *ALAL-1* expression when tumor and normal samples were compared was observed in the LUSC cohort (P value = 1.762e⁻¹¹), where *ALAL-1* was not identified as frequently

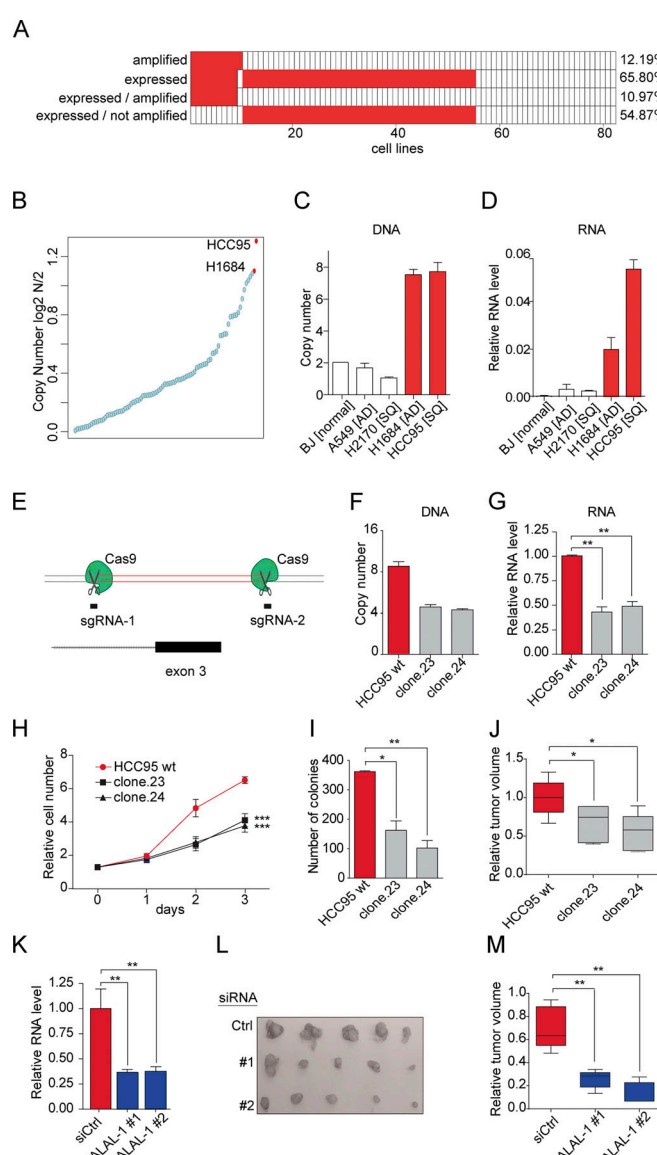

Figure 3. **ALAL-1 promotes an oncogenic phenotype in lung cancer cells.** **(A)** Amplification and expression (FPKM > 0) of *ALAL-1* in lung cancer cell lines. **(B)** Copy number of the *ALAL-1* locus in different lung cancer cell lines. Data were retrieved from the CCLE. **(C)** Experimental quantification of the copy number of the *ALAL-1* locus using qRT-PCR from genomic DNA (gDNA) in different cell lines. AD, adenocarcinoma; BJ [normal], normal human foreskin fibroblasts; SQ, squamous cell carcinoma. **(D)** *ALAL-1* expression in the same cell lines in which copy number was estimated. **(E)** Schematic representation of the CRISPR/Cas9 strategy used to delete exon 3 of ALAL-1. **(F)** Relative DNA copy number of *ALAL-1* exon 3 quantified by qRT-PCR using gDNA from CRISPR/Cas9 clones 23 and 24. **(G)** Expression of *ALAL-1* in clones 23 and 24. RNA levels are represented relative to *ALAL-1* expression in HCC95 cells. **(H)** Cell proliferation determined by MTS assay. **(I)** Colony formation assay in cell lines in which the copy number of *ALAL-1* was reduced. The number of colonies obtained in each condition is represented. **(J)** Volume of tumors obtained after subcutaneous injection of HCC95 cells engineered with the CRISPR/Cas9 technology in immune-compromised mice. **(K)** Knockdown efficiency determined by qRT-PCR in HCC95 cells transfected with control siRNA or ALAL-1–targeting siRNAs (1 and 2). **(L)** Volumes of tumors formed by subcutaneous injection of HCC95 cells in mice after ALAL-1 knockdown. **(M)** Volumes of tumors at the indicated time points in the three experimental conditions tested. Boxplot of *n* = 5. Significance was determined by two-tailed Mann-Whitney test. Bottom and top of the box are the 25th and 75th

percentile (the lower Q1 and upper quartiles, respectively Q3), and the band near the middle of the box corresponds to the median. The lower whisker extends Q1 – 1.5 * interquartile range (IQR) and the upper one Q3 + 1.5 IQR. **(G–K)** Graphs of mean (± SEM) for three independent experiments are shown. Significance was determined by two-tailed unpaired *t* test and represented as *, P ≤ 0.05; **, P ≤ 0.01. Ctrl, control; siALAL-1, ALAL-1 siRNA; siCtrl, control siRNA; FPKM, fragments per kilobase of exon model per million reads mapped; wt, wild type.

amplified. These data suggest that other mechanisms besides gene amplification could be regulating the levels of *ALAL-1*.

DNA methylation changes are common in various types of cancers, and altered methylation associates with changes in gene expression. We analyzed the DNA methylation of the *ALAL-1* locus in lung cancer (Sandoval et al., 2013) and found two differentially methylated CpGs mapping to the 5′ end of *ALAL-1* (both cg26394282, P value = $1.89e^{-24}$ in LUAD and cg16230352, P value = $3.05e^{-19}$ in LUSC; Fig. 2, F–H), suggesting that hypomethylation of the *ALAL-1* gene in tumors could explain the observed higher level of the lncRNA in those tumors that do not present amplification of the locus.

Together, these data indicate that *ALAL-1* is overexpressed in cancer, targeted by genetic and epigenetic mechanisms, with a potential role as an oncogene in NSCLC pathogenesis.

### ALAL-1 promotes the oncogenic phenotype of lung cancer cells

To experimentally test the potential oncogenic role of *ALAL-1*, we set out to identify lung cancer cell lines with a genetic background similar to the one present in the tumor samples (i.e., amplification of *ALAL-1*). For this, we interrogated The Cancer Cell Line Encyclopedia (CCLE), in which around 12% of the lung cancer cell lines bear amplification of *ALAL-1* (Fig. 3 A). Among them, HCC95 (LUSC) and H1648 (LUAD) cell lines showed the highest level of amplification of the *ALAL-1* locus (Fig. 3 B), which we independently estimated as eight copies per cell (Fig. 3 B). Moreover, the level of ALAL-1 RNA was also increased, correlative to the amplification of the gene, when compared with nontumoral cells (BJ) or with lung cancer cell lines without *ALAL-1* amplification (A549 and H2170; Fig. 3, C and D).

To investigate the role of *ALAL-1* amplification in cancer cells, we reverted its genomic amplification by CRISPR/Cas9 genome editing. For this, HCC95 cells were edited by CRISPR using two single guide RNAs (sgRNAs) flanking exon three of ALAL-1, obtaining a deletion of ∼500 bp in the gene (Fig. 3 E and Fig. S2 A). All the cell clones recovered with the intended deletion were heterozygous (Fig. 3 F and Fig. S2 B), probably due to the presence of multiple copies of the gene and to the low frequency of multiple editing events occurring in the same cell. Nevertheless, the obtained clones had a concomitant reduction of ALAL-1 RNA levels (Fig. 3 G). Interestingly, these clones presented reduced cell proliferation (Fig. 3 H), reduced colony formation capacity, and increased apoptosis (Fig. 3 I; and Fig. S2, C and D), as well as reduced tumor growth capacity in xenograft mouse models (Fig. 3 J).

While the genomic deletion of *ALAL-1* indicates a role of the gene in cancer cells, it does not allow determining whether the

observed effect is due to the removal of the DNA sequence or to the decrease of ALAL-1 RNA levels. To uncouple the function of the lncRNA from that of underlying genomic elements, we performed RNAi knockdown experiments with two different siRNAs targeting ALAL-1, which reduced the RNA levels of ALAL-1, leaving the gene locus intact (Fig. 3 K). Similar to what was observed with the ALAL-1 CRISPR clones, depletion of ALAL-1 by RNAi reduced the proliferative and colony formation capacity of HCC95 cells while causing more cell death (Fig. S2, E–G), as well as impaired in vivo tumor formation capacity (Fig. 3, L and M). Comparable effects were observed when ALAL-1 was depleted by RNAi in H1648 cells (Fig. S2, H–J). Conversely, overexpression of ALAL-1 in HCC95, H1648, and A549 cells increased their clonogenic capacity (Fig. S2, K–S) and increased the tumor volumes formed in mice injected with ALAL-1 A549 overxpression cells (Fig. S2 T).

Taken together, our data indicate that ALAL-1 is a functional lncRNA with a pro-oncogenic role in lung cancer.

## ALAL-1 is a cytoplasmic lncRNA that physically and functionally interacts with *squamous cell carcinoma antigen recognized by T cells 3* (SART3)

To further understand ALAL-1 cellular function, we investigated its transcriptional regulation and cellular localization. Analysis of the *ALAL-1* genomic locus identified p65/RelA (RelA Proto-Oncogene, NF-KB Subunit) consensus binding sites around its TSS (Fig. 4 A). Consistently, we observed the presence of ChIP sequencing (ChIP-seq) peaks corresponding to p65/RelA in different cell types (human umbilical vein endothelial cells, Brown et al., 2014; A549, Raskatov et al., 2012; and IMR90, Jin et al., 2013; Fig. 4 A). The association of p65/RelA was detected upon TNF treatment (Fig. 4 A), similar to other nuclear factor κB (NF-κB) bona-fide target genes (Fig. S3, A–D). Also, in agreement with this, the level of ALAL-1 was reduced when p65/RelA was depleted (Fig. 4 B), while it was induced by TNF treatment (Fig. 4, C and D). Moreover, the 5′ genomic region of *ALAL-1* (959 bp) containing the NF-kB binding motif was able to drive the activation of a luciferase gene following TNFα treatment when cloned into a reporter vector (Fig. S3 E). Together, these data demonstrate that *ALAL-1* is a transcriptional target of NF-κB.

Of note, *ALAL-1* is localized upstream of *IKBKB*, a known NF-κB target gene (Fig. S4 A). While *IKBKB* is coamplified with *ALAL-1* in some tumors due to their genomic proximity (Fig. S4 B), we did not observe any effect on *IKBKB* expression upon *ALAL-1* truncation by CRISPR or by siRNA knockdown (Fig. S4, C–F). Depletion of *IKBKB* by RNAi instead reduced the level of ALAL-1 (Fig. S4 G), in line with the notion that *ALAL-1* is a transcriptional target of NF-κB. These data suggest that although both genes are coregulated by NF-κB, at least the effects observed on cancer phenotypes upon ALAL-1 depletion are not driven by *IKBKB*.

To localize ALAL-1 in the cell, we performed subcellular fractionation and RNA FISH experiments. The results showed that ALAL-1 predominantly localizes to the cytoplasm in lung cancer cells (Fig. 4, E and F). Quantification by independent methods (i.e., qRT-PCR and RNA FISH) estimated that HCC95 cells express on average >150 molecules of ALAL-1 per cell

(Fig. 4 G). We therefore investigated the possibility that ALAL-1 could be acting in the cytoplasmic compartment through interaction with proteins. For this, we performed in vitro RNA pull-down experiments combined with mass spectrometry. The results showed that the protein that specifically interacts with ALAL-1 with the highest number of peptides is SART3 (Fig. 5 A). This interaction was confirmed in independent experiments, with observation of either endogenous or myc-tagged transfected SART3 binding to ALAL-1, while no SART3 binding was observed when a control antisense ALAL-1 RNA was used (Fig. 5 B). Moreover, the interaction between ALAL-1 and SART3 was confirmed using RNA immunoprecipitation (RIP; Fig. 5 C). U6 RNA, a known SART3 interactor (Bell et al., 2002), was used as a positive control, showing high enrichment in the SART3 immunoprecipitates, while no enrichment was observed for the abundant MALAT1 lncRNA (Fig. 5 C).

To further investigate the interaction between ALAL-1 and SART3, we tested whether the ability of ALAL-1 to bind the protein was dependent on the SART3 putative binding sites present in the lncRNA sequence (Liu et al., 2015a). For this, RNAs corresponding to full-length or different fragments of ALAL-1 were obtained by in vitro transcription, and their interaction with SART3 was tested by RNA pulldown and Western blot (Fig. 5 D). As observed, the only two ALAL-1 fragments able to pull down SART3 were fragments #3 and #6, containing two SART3 putative binding sites located within the ALAL-1 195–250 nt region (Fig. 5 D), which overlaps part of the ALAL-1 CRISPR deletion (Fig. S5, A–C). While the enforced expression of full-length ALAL-1 resulted in increased clonogenic capacity of cells with ALAL-1 deletion, the overexpression of SART3 alone, interacting or not with ALAL-1 fragments, did not have this effect (Fig. S5 D). These results suggest that the central part of the ALAL-1 sequence is responsible for the specific interaction between ALAL-1 and SART3, which is required for lncRNA function; however, ALAL-1 truncations are not sufficient to induce a phenotype of increased cell proliferation, suggesting that additional regions of the lncRNA may be necessary for ALAL-1 cellular activity.

To further confirm the functional role of ALAL-1 in trans in cooperation with SART3, we overexpressed ALAL-1 in cells depleted of the lncRNA by CRISPR/Cas9 (i.e., clones 23 and 24). As expected, we observed an increase in the number of cell colonies (Fig. S5, E and F). However, the increased colony formation capacity driven by ALAL-1 overexpression was abolished when SART3 was inhibited (Fig. S5 D; and S5, G–I). Taken together, these results suggest dependence on SART3 for ALAL-1 function.

## ALAL-1 regulates USP4 subcellular localization and function

Several functions have been reported for SART3. While it is known to act in the nucleus as a U6 recycling factor during splicing (Bell et al., 2002), it is also found in the cytoplasm of proliferating cells (Sasatomi et al., 2002; Suefuji et al., 2001), including the lung cancer cell lines used in this study (Fig. S5 J). SART3 has been shown to interact with the ubiquitin-specific proteases USP4 and USP15 (Park et al., 2016; Zhang et al., 2016), which are known to modulate the Wnt/β-catenin, NF-κB, p53, and TGF-β signaling pathways (Fan et al., 2011; Eichhorn et al.,

**Figure 4.  *ALAL-1 is a transcriptional target of NF-κB.* (A)** ChIP-seq signal of p65 in the ALAL-1 locus in HUVEC cells treated or not with TNFα. Peaks called in p65 ChIP-seq analysis in the A549 cell line and the consensus sequences corresponding to p65 binding sites are indicated below. In green, annotated ALAL-1 isoforms. **(B)** ALAL-1 RNA levels and p65 protein levels determined in HCC95 cells after p65 knockdown. Graphs of mean (± SEM) for three independent experiments are shown. Significance was determined by two-tailed unpaired *t* test and represented as **, P ≤ 0.01; ***, P ≤ 0.001. **(C)** Time course experiment showing the induction of *ALAL-1* in HCC95 cells treated with TNFα. **(D)** RNA-seq tracks of *ALAL-1* locus of HCC95 cells untreated or treated with TNFα for 4 h. In green, ALAL-1 annotated isoforms; in red, ALAL-1 isoform identified and studied. **(E)** Subcellular localization of ALAL-1 assessed by cell fractionation in the indicated cell lines. **(F)** RNA fluorescence in situ hybridization in HCC95 cells untreated and treated with TNFα. **(G)** Number of fluorescent foci detected by RNA FISH in cells transfected with control siRNA (siCtrl) or ALAL-1 siRNA (siALAL-1). At least 100 cells per condition were counted. Graph of mean (± SEM) for three independent experiments is shown. Significance was determined by two-tailed unpaired *t* test and represented as *, P ≤ 0.05; ***, P ≤ 0.001. HUVEC, human umbilical vein endothelial cells; RelA, RelA proto-oncogene, NF-ΚB subunit; siCtrl, control siRNA; sip65, p65-targeting siRNA; Nuc, nucleus; Cyt, cytoplasm.

2012; Zhang et al., 2012, 2011; Zhao et al., 2009). We observed that ALAL-1 inhibition did not affect SART3 mRNA or protein levels (Fig. S5 K). To understand the functional relationship between ALAL-1 and SART3, we knocked down one or the other and performed RNA-seq to identify gene expression changes in untreated or TNF-treated HCC95 cells. Three independent experiments were performed for each condition, and RNA depletion was confirmed by qRT-PCR prior to sequencing (Fig. S5 L). Depletion of ALAL-1 in untreated cells affected the expression of

1,905 genes (adjusted [adj.] P value < 0.05), while in TNF-treated cells, 1,766 genes were altered (Fig. S5 M and Table S2). On the other hand, depletion of SART3 in untreated cells affected the expression of 1,082 genes (adj. P < 0.05), while in TNF-treated cells, 517 genes were altered (Fig. S5 M and Table S2).

For both ALAL-1– and SART3-depleted cells, gene ontology (GO) analysis identified a significant enrichment in biofunctional terms related to decreased cell migration, proliferation, and survival (Fig. 5 E), consistent with a functional role of ALAL-1

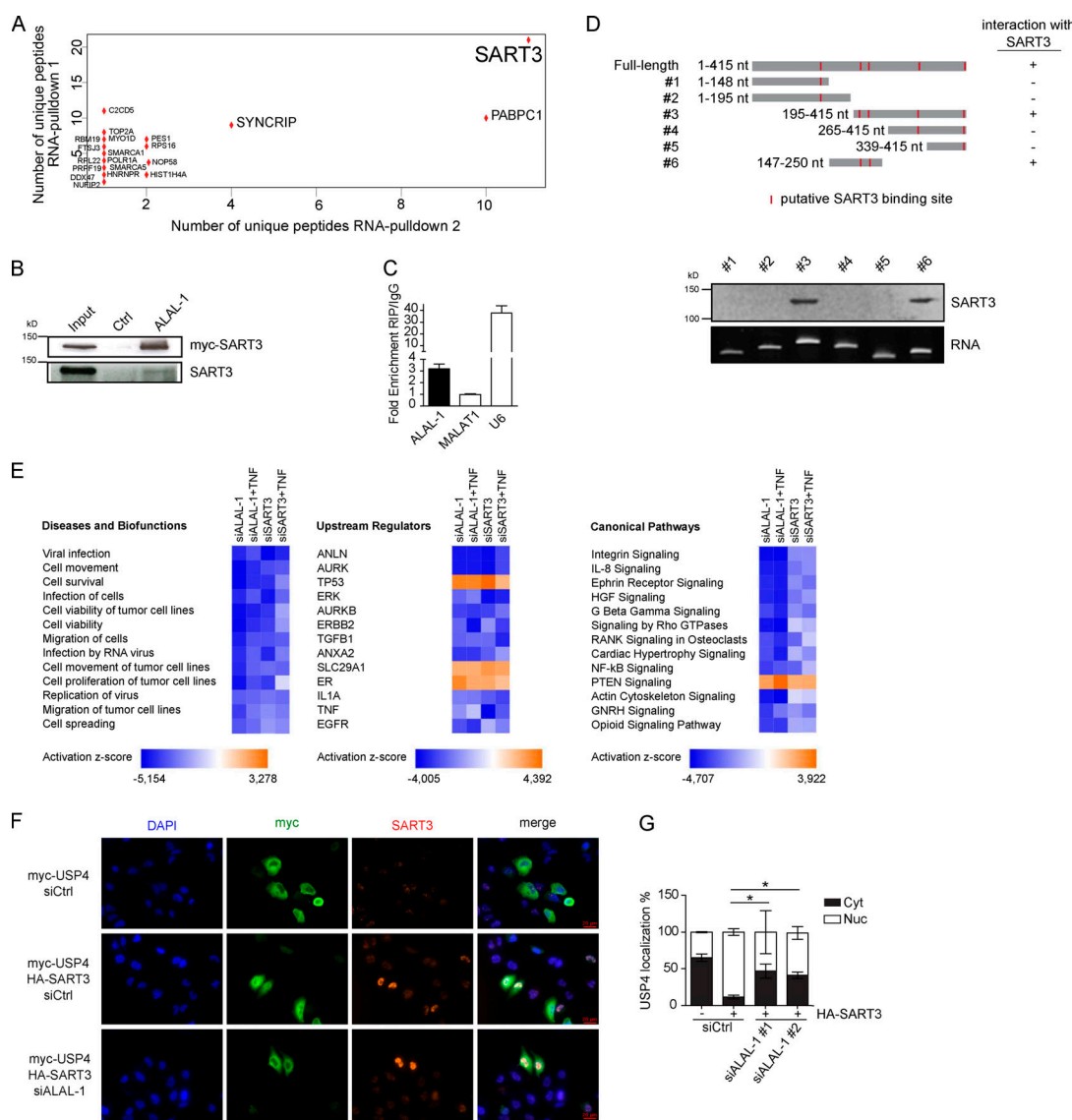

Figure 5. **ALAL-1 interacts with SART3 and regulates key cellular pathways. (A)** Dot plot indicating the number of peptides detected of each of the proteins identified as bound to ALAL-1 but not the control RNA in two independent RNA-pulldown experiments. **(B)** SART3 detection by Western blot in independent ALAL-1 RNA-pulldown experiments. **(C)** RIP of SART3 followed by qRT-PCR of the bound RNAs. Values are represented relative to the IgG control. **(D)** SART3 detection by Western blot on RNA-pulldown of different ALAL-1 fragments represented schematically on the right. The localization of putative SART3 binding motifs (ACAGA) is indicated with red lines. **(E)** Diseases and biofunctions, upstream regulators, and canonical pathways significantly affected in HCC95 cells upon ALAL-1 or SART3 depletion by siRNA knockdown relative to corresponding control siRNA condition. **(F)** Immunofluorescence showing SART3 (anti–SART3 ab) and USP4 (anti–myc ab) localization in the indicated conditions. **(G)** Quantification of the localization of USP4 in the indicated conditions. Graphs of mean (± SD) for three independent experiments, 15 images per condition, are shown. Significance was determined by two-tailed unpaired *t* test and represented as *, P ≤ 0.05. Ctrl, control; siALAL-1, ALAL-1 siRNA; siCtrl, control siRNA; siSART3, SART3-targeting siRNA; HGF, hepatocyte growth factor; RANK, receptor activator of NF-kB; PTEN, phosphatase and tensin homolog; GNRH, gonadotropin-releasing hormone; ab, antibody; Cyt, cytoplasm; Nuc, nucleus.

in these processes and in agreement with the phenotype observed. Moreover, in cells depleted of ALAL-1, among the most represented upstream regulators and canonical pathways called by the GO analysis, we identified TNF, p53, NF-κB, TGF-β1, and interleukin (IL)-8 (Fig. 5 E). Consistent with a functional interaction between ALAL-1 and SART3, the same regulators and pathways were found to be significantly affected when SART3 was depleted (Fig. 5 E), suggesting that the lncRNA acts in coordination with this protein. Interestingly, the pathways found to be affected by either ALAL-1 or SART3 depletion were

previously reported in different studies to be modulated by SART3 directly or indirectly through its interacting partner USP4 (Fan et al., 2011; Li et al., 2016; Xiao et al., 2012; Zhang et al., 2011), suggesting that the interaction of ALAL-1 with SART3 could be regulating the SART3–USP4 complex.

USP4 is a key regulator of the NF-κB pathway among others (Fan et al., 2011; Zhang et al., 2011; Zhao et al., 2009), able to deubiquitinate upstream factors of the pathway such as TAK1, TRAF2, and TRAF6, which are polyubiquitinated upon TNF treatment (Fan et al., 2011; Xiao et al., 2012). Considering the

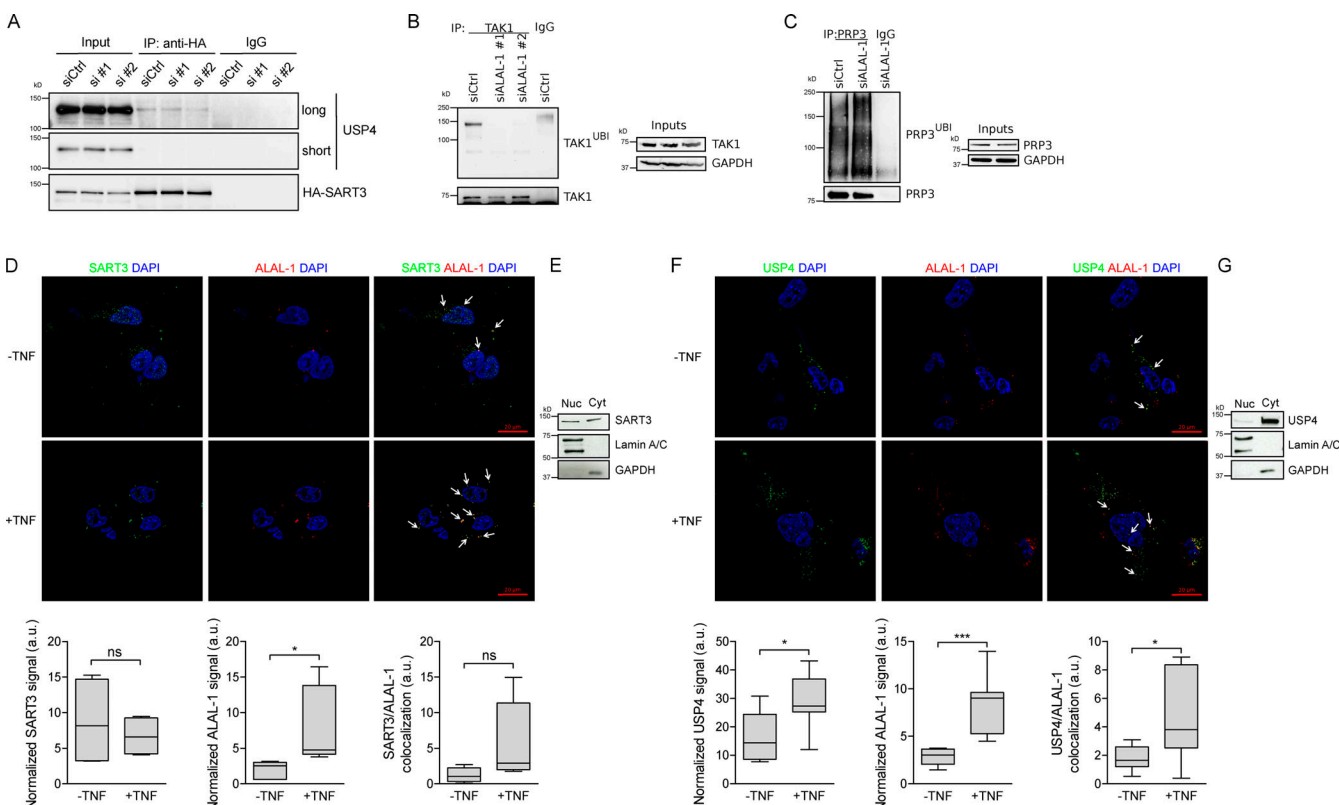

Figure 6. **ALAL-1 interactions regulate USP4 localization and deubiquitinase activity. (A)** Co-immunoprecipitation of USP4 and SART3 in HCC95 transfected with HA-SART3 and siRNAs targeting ALAL-1. **(B and C)** Ubiquitination of TAK1 and PRP3 in cells depleted or not of ALAL-1 by RNAi evaluated by immunoprecipitation and Western blot. **(D)** Co-immuno–FISH of endogenous SART3 and ALAL-1 with and without TNF treatment. Graphs show SART3, ALAL-1, and colocalization signal normalized to DAPI. Graphs of median (± SD) for three independent experiments are shown. Significance was determined by two-tailed unpaired *t* test and represented as *, P ≤ 0.05. **(E)** SART3 localization evaluated by Western blot of cell fractions (Nucleus ([Nuc]/Cytoplasm [Cyt]). **(F)** Co-immuno–FISH of endogenous USP4 and ALAL-1 with and without TNF treatment. Graphs show USP4, ALAL-1, and colocalization signal normalized to DAPI. Graphs of median (± SD) for three independent experiments are shown. Significance was determined by two-tailed unpaired *t* test and represented as *, P ≤ 0.05; ***, P ≤ 0.001. **(G)** USP4 localization evaluated by Western blot of cell fractions (Nucleus/Cytoplasm). ns, not significant; siALAL-1, ALAL-1 siRNA; siCtrl, control siRNA; IP, immunoprecipitation; Nuc, nucleus; Cyt, cytoplasm.

pathways affected by ALAL-1 and SART3 depletion and the reported role for SART3 in the regulation of USP4 (Park et al., 2016), we hypothesized that the interaction between ALAL-1 and SART3 could affect USP4 function and, as a consequence, the signaling pathways that regulate the inflammatory response. We therefore analyzed USP4 shuttling between the cytoplasm and the nucleus, which is known to be regulated by SART3 (Song et al., 2010; Zhang et al., 2016). As previously described, over-expression of SART3 resulted in nuclear translocation of USP4 (Fig. 5, F and G). However, when ALAL-1 was depleted, the SART3-mediated nuclear translocation of USP4 was partially impaired (Fig. 5, F and G), while the interaction between SART3 and USP4 was not affected by depletion of ALAL-1 (Fig. 6 A), indicating that ALAL-1 is involved in USP4 relocalization without impairing SART3-USP4 interaction in bulk.

Our results suggest that ALAL-1 could be affecting the cellular function of USP4. To evaluate the consequence of ALAL-1–dependent USP4 localization, we analyzed the deubiquitination ability of USP4 on its substrates in the presence or absence of ALAL-1. For this, we selected the cytoplasmic protein TAK1, a known substrate of USP4 (Fan et al., 2011) and key regulator of the NF-κB pathway (Fan et al., 2011; Xiao et al., 2012).

Interestingly, the amount of ubiquitinated TAK1 was reduced upon ALAL-1 depletion (Fig. 6 B). Conversely, the ubiquitination levels of PRP3 (Song et al., 2010), a nuclear substrate of USP4, were increased upon ALAL-1 depletion (Fig. 6 C). These results are in agreement with the role of ALAL-1 in inducing USP4 nuclear localization.

To have a better understanding of the interaction between ALAL-1 and SART3 and USP4, we performed coimmuno-FISH experiments coupled with confocal microscopy. SART3 localized in the cytoplasm and to a less extent in the nucleus, as observed by microscopy (Fig. 6 D) and Western blot (Fig. 6 E). ALAL-1 predominantly localized in the cytoplasm, and its signal significantly increased with TNF treatment (Fig. 6 D). Coimmuno-FISH analysis showed clear colocalization of SART3 and ALAL-1 in the cytoplasm (Fig. 6 D). Similarly, ALAL-1 clearly colocalized with USP4 in this cellular compartment (Fig. 6, F and G). For both SART3 and USP4, the colocalization with ALAL-1 was higher following TNF treatment (Fig. 6, D and F), probably due to higher expression levels of ALAL-1 following TNF treatment, as suggested by normalizing the colocalization signal to the signal of ALAL-1 (Fig. S5 N). Altogether, the data suggest that ALAL-1, SART3, and USP4 preferentially colocalize

in the cytoplasm, which somehow triggers increased levels of USP4 nuclear translocation. As a result, there is a switch from cytoplasmic to nuclear USP4 substrates with subsequent changes in their ubiquitination levels.

**ALAL-1 induces immune evasion of non–small cell lung tumors**

Interestingly, several genes affected by either ALAL-1 or SART3 depletion encode for inflammatory factors or are components of NF-κB and IL-8 signaling pathways, regulating inflammatory mediators (Fig. 5 E). Moreover, USP4 targets TAK1 for deubi-quitination, which is a key regulator of the NF-κB pathway. Taking into consideration the critical role of these pathways in the interaction between the tumor and its microenvironment in cancer progression, we decided to investigate the relationship between ALAL-1 and the immune environment of the tumor. To determine the relationship between ALAL-1 expression and the level of tumor infiltration by immune populations, TCGA LUSC samples were grouped in six subgroups ranging from low to high presence of cytotoxic cells based on the enrichment of gene signatures of specific immune cell populations (Tamborero et al., 2018; Fig. 7 A), and ALAL-1 expression in each of these sub-groups was assessed. Interestingly, the expression of ALAL-1 was significantly elevated in tumors with lower levels of im-mune infiltration, while the tumors with the highest levels of immune infiltration expressed ALAL-1 at lower levels (Fig. 7 A). To confirm this observation, we quantified the expression of ALAL-1 in an independent cohort of LUSC patients and deter-mined the presence of PD-1–positive infiltrating cells, which is related to the strength of T cell receptor signaling and thus to the functional avidity of specific T cells (Simon and Labarriere, 2017). In agreement with our previous data, tumors with lower levels of PD-1–positive cells expressed higher levels of ALAL-1 and vice versa (Fig. 7 B).

Next, to explore the relationship between ALAL-1 and spe-cific subtypes of immune cells in lung adenocarcinomas, we classified TCGA LUAD tumors based on the amplification state of the *ALAL-1* gene and determined the enrichment of cell type–specific gene signatures (Hänzelmann et al., 2013; Tamborero et al., 2018). Compared with tumors without *ALAL-1* amplifica-tion, lung adenocarcinomas with amplified *ALAL-1* presented significantly lower levels of several immune populations, such as T memory, T follicular helper, and dendritic cells (Fig. 7 C). These observations suggest that the increased expression of ALAL-1 concurs with a decreased level of immune infiltration.

Elevated levels of ALAL-1 in tumor cells may cause alterations of signaling pathways that regulate inflammatory mediators, leading to an altered inflammatory response. To investigate this hypothesis, we quantified the level of cytokines secreted to the media by A549 overexpressing ALAL-1 cells or A549 cells transduced with an empty vector as control. This analysis showed a strong decrease in the concentration of several of these molecules in the culture media of ALAL-1 A549 overexpressing cells, including CXCL1, IL-6, and CXCL10 (Fig. 7 D). Then, we experimentally tested if the increased expression of the lncRNA could influence the attraction immune populations by these cells. For that, we cultured ALAL-1 overexpressing or con-trol A549 cells and assayed their effect on the migration of

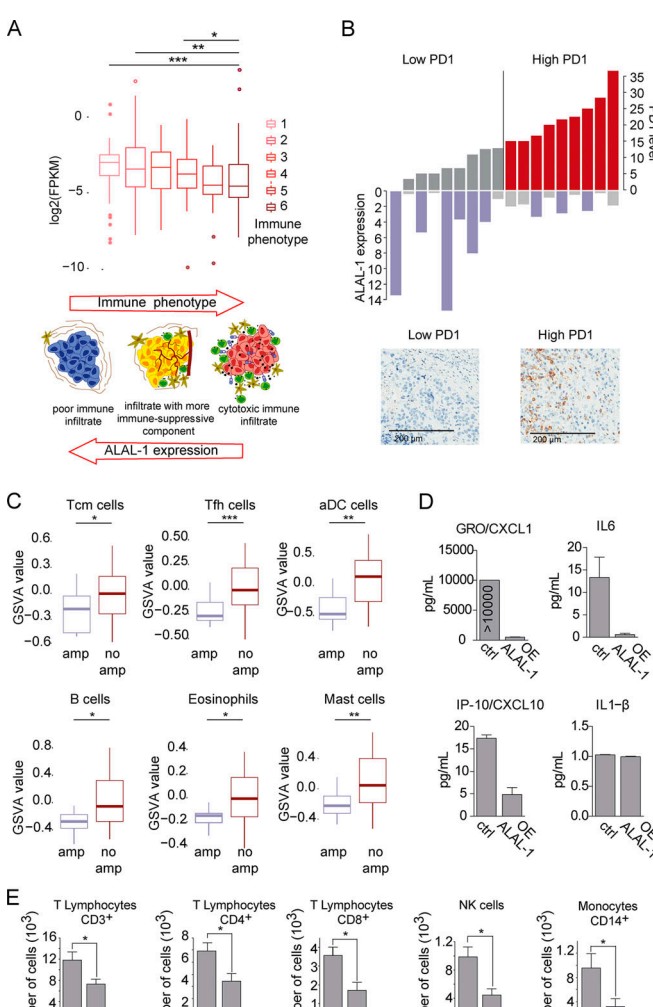

Figure 7. **ALAL-1 contributes to the immune evasion of lung tumors.**
**(A)** Expression of ALAL-1 in TCGA-LUSC cohort, classified based on different immune phenotypes based on gene expression signatures of different infil-tration patterns and levels of cytotoxic cells (Tamborero et al., 2018). Schematic representation adapted from Tamborero et al. (2018). Significance was determined by two-tailed unpaired Wilcoxon test. **(B)** ALAL-1 expression and level of PD-1–positive cells in an independent cohort of LUSC tumors. **(C)** Level of infiltration of the indicated cell populations computed as Gene Set Variation Analysis (GSVA) score in the TCGA-LUAD cohort of tumors with (n = 39) or without (n = 320) *ALAL-1* gene amplification. Significance was determined by two-tailed unpaired Wilcoxon test. **(D)** Level of cytokines secreted to the media of A549 cells that overexpress ALAL-1 asquantified by Luminex assay. **(E)** Number of cells of different subpopulations from pe-ripheral blood migrated to A549 cells overexpressing ALAL-1 or A549 control cells. Graphs of mean (± SEM) for three independent experiments are shown. Significance was determined by two-tailed unpaired *t* test and represented as *, P ≤ 0.05; **, P ≤ 0.01; ***, P ≤ 0.001. FPKM, fragments per kilobase of exon model per million reads mapped; Tcm, T cell memory; Tfh, T follicular helper cells; aDC, artificial dendritic cells; amp, amplification; GRO, chemokine (C-X-C motif) ligand 1; OE, overexpression; IP-10, C-X-C motif chemokine ligand 10; NK, natural killer; ctrl, control.

peripheral blood cells (Fig. 7 E). In concordance with changes in the composition of secreted factors, the enforced expression of ALAL-1 resulted in decreased migration toward the tumor cells of several immune cell populations, such as CD8[+], CD3[+], and

CD4$^+$ lymphocytes, natural killer cells, and monocytes (Fig. 7 E), confirming an inhibitory effect of ALAL-1 in their capacity to chemo-attract cells. Together, our results suggest that by influencing the levels of pro-tumoral inflammatory mediators in the microenvironment, ALAL-1 reduces the infiltration by immune populations favoring tumor progression.

## Discussion

In the past, the identification of cancer-associated genes mainly focused on the analysis of the genetic alterations that target protein-coding genes. However, restricting the studies to coding genes misses the opportunity to discover cancer drivers that reside in the noncoding part of the genome. Here, by integrating copy number alteration and gene expression data analyses from thousands of tumors, we identified a number of genomic regions that contain lncRNAs with potential roles in cancer. Our findings represent a relevant extension of previous works on SCNAs, since they rely on a higher number of tumors and types of cancers analyzed (Akrami et al., 2013; Hu et al., 2014; Yan et al., 2015). Moreover, our astringent approach only considers focal peaks that exclusively map to lncRNAs, which helps the unequivocal ascription of the function to the noncoding loci. Although this strategy may result in the loss of lncRNAs that act in coordination with other gene loci, it led us to identify ALAL-1, as an lncRNA with oncogenic features.

The involvement of lncRNAs in the execution of cellular programs has been ascribed to three different levels of gene activity: (i) the underlying genomic sequence of the locus, which contains elements able to bind regulatory proteins such as transcription factors; (ii) the act of transcription, which can either act as a positive feedback or cause transcriptional interference; and (iii) the RNA product itself (Marchese et al., 2017). Our experimental data, which represent the combination of several orthogonal methods, are quite compelling in demonstrating that the oncogenic role of ALAL-1 is dependent on its RNA product. Although the identified focal peak solely maps to ALAL-1, the amplified region usually spans a large chromosomal fragment that comprises several gene loci. We cannot therefore exclude that functional interactions exist between ALAL-1 and other coamplified genes. Still, our data strongly support that ALAL-1 per se has a pro-oncogenic effect. Moreover, while the initial identification of ALAL-1 was due to its frequent amplification in lung adenocarcinomas, the lncRNA is also overexpressed in squamous tumors (head and neck and lung), indicating that ALAL-1 also plays a prominent role in squamous cancer.

ALAL-1 is transcriptionally activated by NF-κB and is a functional component of this signaling axis. This important pathway is finely tuned at many levels, with some lncRNAs shown to be involved in its regulation (Rapicavoli et al., 2013; Liu et al., 2015b; Chen et al., 2017). In contrast to several other functional lncRNAs, ALAL-1 is remarkably enriched in the cell cytoplasm, where it interacts with the RNA binding protein SART3. By interacting with SART3, ALAL-1 can modulate a specific facet of this multifunctional protein. ALAL-1 is able to colocalize in the cytoplasm with both SART3 and USP4.

Moreover, knockdown of ALAL-1 diminishes the localization of USP4 into the nucleus mediated by SART3, suggesting that the presence of ALAL-1 favors the nuclear import of USP4. It is possible that ALAL-1 induces conformational changes in SART3/USP4 that promote their translocation, for which importin has been shown to be required (Park et al., 2016). Although structural studies are needed to shed light on the mechanistic details of this phenomenon, it is clear that the transcriptional activation of ALAL-1 in response to TNF and its subsequent association with SART3 allows coordination between these factors to induce an increased nuclear localization of USP4. As a result, there is a switch from cytoplasmic to nuclear USP4 substrates with functional consequences. Indeed, the gene expression changes common to SART3 and ALAL-1 knockdown could be the result of the alteration of regulatory activity of the USP4-dependent ubiquitination level of proteins.

Due to ALAL-1 impacts on cancer cells, the lncRNA emerges as a so-far unknown modulator of the tumor microenvironment. Data from patient-derived samples show that the in vivo expression of ALAL-1 is inversely correlated with the high immune infiltration of LUSCs. This suggests that ALAL-1–overexpressing tumors are less susceptible to the anti-tumor response displayed by the immune system. Our observations therefore point to ALAL-1 as a possible target for lung cancer therapies, suggesting that the in vivo inhibition of ALAL-1 could have a "double-hit" anti-tumor effect: on one hand, by decreasing the autonomous capacity of cells to survive and proliferate and, on the other hand, by promoting immune infiltration and response against the tumor. Nowadays, treatments with PD-L1 and PD-1 blockers, intended to induce anti-tumor lymphocytes, fail in 80% of NSCLC patients (Borghaei et al., 2015; Horn et al., 2017). The resistance is thought to be due to several mechanisms that lead to innate evasion, in which T cells and other immune cells are excluded from the tumor microenvironment (Spranger and Gajewski, 2018). We thus speculate that the combination of immune checkpoint blockers with ALAL-1 inhibitors may represent an opportunity to treat patients with disease that is otherwise refractory to these types of treatments. Rapid progress in RNA-targeting therapeutics raises hopes of bringing this application closer to the clinic.

## Materials and methods
### Patients
Copy number tumor data were retrieved from several previously published datasets, including TCGA data for 7,448 tumors across 25 cancer types downloaded from the Firebrowser server (2013_10), 85 LUAD tumors from CIMA-CUN, Gene Expression Omnibus (GEO) accession no. GSE72195 (Aramburu et al., 2015), and 162 from The University of Texas MD Anderson Cancer Center, accession no. GSE72195 (Aramburu et al., 2015), as well as 101 from Uppsala University, accession no. GSE28582 (Micke et al., 2011). The validation set of LUSC patients was obtained from the Centro de Investigación Biomédica en Red de Enfermedades Respiratorias (CIBERES) multi-institutional Pulmonary Biobank Platform (Madrid, Spain).

## Data processing

TCGA downloaded data contain the results of copy number alterations obtained from the analysis of Affymetrix 6.0 single nucleotide polymorphism arrays using the GISTIC algorithm. Using in-house R and Perl scripts, copy number alterations were annotated (GENCODEv19) and classified (biotypes). The expression data were retrieved from MiTranscriptome and TCGA. From additional lung cancer cohorts (accession nos. GSE18842, GSE19804, and GSE19188) expression of *ALAL-1* was obtained with the probe 231378_at. Methylation data (HumanMethylation 450K BeadChip) were retrieved from TCGA Wanderer interface (Díez-Villanueva et al., 2015). In addition CCLE resources (http://www.broadinstitute.org/ccle) were used to assess *ALAL-1* expression in cancer cell lines. Enrichment analysis of GOs from the differentially expressed genes was obtained with the R package clusterProfiler (Yu et al., 2012). Ingenuity Pathway Analysis was used for additional data interpretation.

## CRISPR/Cas9 editing

CRISPR/Cas9 sgRNAs were designed with the tool available at http://crispr.mit.edu/. To delete exon 3 of *ALAL-1*, the two sgRNAs with the highest score were cloned in pX330 as described in the Zhang laboratory CRISPR protocol. The sequences of the sgRNAs used are listed in Table S3. To generate the genomic deletion, HCC95 cells were cotransfected with pX330–sgRNA-1 and pX330–sgRNA-2 and GFP-expressing plasmid. Control cells were transfected with pX330 vector lacking any sgRNA. 24 h after transfection, GFP-positive cells were sorted in six 96-well plates using the BD FACSAria IIu cytometer. Cells were left to grow until they reached confluency. Genomic DNA was then extracted with the QuickExtract reagent (Epicentre). Genotyping was performed with PCR primers upstream and downstream of the sgRNA cleavage sites (Table S3). PCR products were then run in an agarose gel to check for amplicon size. PCR products of the clones carrying the deletion were sent for Sanger sequencing.

## Clonogenicity, cell proliferation, and apoptosis assays

Approximately 500 cells were plated in each well of a six-well plate. After a period of 10–14 d, cells were fixed using 0.5% glutaraldehyde and stained with 1% crystal violet in 35% methanol. Colonies counting was performed manually. Crystal violet staining was solubilized with 10% acetic acid, and absorbance was measured at 570 nm on a spectrophotometer plate reader. For cell proliferation assays, $10^3$ HCC95, $1.5 \times 10^3$ H1648, or 500 A549 cells were plated in each well of a 96-well plate, and proliferation was assessed using the CellTiter Aqueous Non-Radioactive Cell Proliferation Assay kit (MTS; Promega). Apoptosis was measured by flow cytometry using the Annexin V and 7-amino-actinomycin D staining kit (BD Biosciences) and FlowJo analysis software.

## Cell culture, RNAi, and TNFα treatment

All the cell lines used (BJ, A549, NCI-H2170, NCI-H1648, and HCC95) were cultured at 37°C in the presence of 5% $CO_2$ using either DMEM or RPMI 1640 medium (GIBCO) supplemented with 10% fetal bovine serum and penicillin–streptomycin (1%). Short tandem repeat profiling was used to authenticate cell lines, and cells were tested for mycoplasma contamination regularly using the MycoAlert Mycoplasma Detection Kit (Lonza). For *ALAL-1* inhibition, $2 \times 10^5$ cells per well were plated in a six-well plate. The next day, siRNAs at a final concentration of 30 nM siRNA were transfected using Lipofectamine 2000 (Invitrogen) following the manufacturer's protocol. siRNAs are listed in Table S3. TNFα treatment was performed using a concentration of 10 ng/ml of TNFα (R&D Systems; 210-TA) for the indicated time points.

## Promoter reporter assay

The ALAL-1 genomic sequence (∼950 bp) flanking the NF-kB motif was amplified from human genomic DNA and cloned into pGL3-basic vector (Promega) at XhoI and KpnI sites with the following primers: 5′-CACCCTCGAGACTCAGAGCCCCAA ATCCTT-3′ and 5′-CACCGGTACCGTCACTCTCGTGGCCATCTT-3′. The pNF-kB–luc plasmid containing five NF-kB response elements was from Clontech. TK-Renilla plasmid was used as normalizing control. Firefly and Renilla luciferase activities were measured using the dual luciferase reporter assay kit (Promega) and a luminometer.

## Determination of cytokines and chemokines in cell supernatants

Supernatants were collected 24 h after seeding of the cells and centrifuged to remove debris before assay. A human cytokine/chemokine MILLIPLEX MAP kit (Millipore) was used to measure GRO/CXCL1, IL-6, CXCL10, and IL-1β concentrations in supernatants following the manufacturer's protocol. A Luminex 100/200 System was used to run the plate, compute standard curves, and calculate cytokine and chemokine concentrations.

## RNA-seq

Total RNA from HCC95 cells was extracted using the Maxwell 16 Total RNA Purification Kit (Promega). Triplicates for each condition (*ALAL-1* inhibition, TNFα treatment, and controls) were done. RNA quality for each sample was assessed using the Agilent 2200 TapeStation. Library preparation was performed following the MARS-seq protocol (Jaitin et al., 2014). Libraries were then sequenced on an Illumina NextSeq. Sequenced reads were aligned using STAR (against hg19), and differential gene expression analysis was performed with DeSeq2. RNA-seq data are available at the GEO database under accession no. GSE114632.

## DNA extraction and gene copy number estimation

Genomic DNA was obtained from $2 \times 10^5$ cells using a DNA extraction kit (QIAGEN). Copy number was assessed by quantitative PCR (qPCR) using primers recognizing the *ALAL-1* locus. Data were normalized to the *PEX19* gene located in chromosome 1p36.23, a region with no significant aneuploidy in the cancer cell lines studied. Copy number of the *ALAL-1* locus in CRISPR/Cas9 engineered cells was quantified using the same methodology.

## RNA extraction, qPCR, and primer design

Total RNA was extracted using Trizol reagent (Sigma). After DNase I (Invitrogen) treatment, RNA was reverse transcribed with the High Capacity Kit (Applied Biosystems). RT-PCRs were

performed in quadruplicates, and relative gene expression was obtained using *HPRT* as the housekeeping gene. All primers were designed using the Universal Probe Library Assay Design Center (Roche) and are listed in Table S3. To evaluate the absolute number of *ALAL-1* RNA molecules per cell, total RNA was isolated from HCC95 cells accurately counted using a Countess Automated Cell Counter (Thermo Fisher). RNA extraction and cDNA generation were performed as described using 1 µg of RNA. The standard curve was obtained by qPCR of serial dilutions of a known amount of *ALAL-1* RNA in vitro transcribed and used to calculate the copy number of *ALAL-1* per cell.

### RNA pulldown and RIP
RNA pulldown was performed according to Marín-Béjar et al. (2013). Briefly, biotinylated RNA was generated in vitro and incubated with total protein extract of HCC95 cells and then streptavidin magnetic beads. Interacting proteins were loaded in a NuPAGE Novex 4%–12% bis-Tris gel (Invitrogen) and stained with the SilverQuest Silver Staining Kit (Thermo Fisher). For mass spectrometry analysis, differential bands were submitted to the Taplin Mass Spectrometry Facility (Harvard University). For RIP experiments, $10^7$ A549 cells overexpressing *ALAL-1* were lysed with lysis buffer (20 mM Tris-HCl, pH 7.5, 100 mM KCl, 5 mM $MgCl_2$, 0.5% NP-40, protease inhibitors [Roche], RNase inhibitor [100 U/ml], and 10 mM DTT). Protein lysate was then incubated with prewashed protein A magnetic beads for 1 h with rotation at 4°C for preclearing. Extract was diluted up to 1 ml with RIP buffer and incubated either with Normal Rabbit IgG (Cell Signaling; rabbit, 2729S) or with anti-SART3 (Abcam; rabbit, ab155765) overnight with rotation at 4°C. Protein A magnetic beads were added for 1 h and then washed five times with Buffer A (150 mM KCl, 25 mM Tris-HCl, pH 7.4, 5 mM EDTA, 0.5% NP-40, and 0.5 mM DTT); for the last wash, PBS was used. RNA was recovered from the beads with Trizol reagent.

### Immunoprecipitation
ALAL-1 siRNA knockdown was performed in HCC95 cells, and after 2 d, cells were lysed and immunoprecipitated with the appropriate antibody (anti-HA, mouse, sc-7392; anti-PRP3, rabbit, A302-074; or anti-TAK1, rabbit; Thermo Fisher; 700113) and protein G magnetic beads. IgG was used as a control for immunoprecipitation. 10% of the lysate was used as input. Ubiquitinated proteins were detected by immunoblot using a mono- and polyubiquitinylated conjugates monoclonal antibody (FK2, mouse; Enzo; #BML-PW8810).

### *ALAL-1* overexpression
*ALAL-1* cDNA sequence was cloned between *EcoRI-BamHI* sites in pcDNA3.0 vector (Invitrogen) and between *EcoRI-XhoI* sites in pMSCVneo retroviral vector (Clontech) for transient and stable overexpression, respectively. For transient overexpression, plasmids were transfected with Lipofectamine 2000 using the manufacturer's protocol.

### Nuclear-cytoplasmic fractionation
$3 × 10^6$ cells were lysed in 500 µl of lysis buffer (20 mM Tris-HCl, pH 7.5, 0.1% NP-40, 280 mM NaCl, 3 mM $MgCl_2$, and

RNasin; Promega) and incubated on ice for 10 min. The cell lysate was then layered over 500 µl of sucrose cushion (50% sucrose in cell lysis buffer) and centrifuged at 13,000 rpm at 4°C for 10 min. The resulting supernatant corresponded to the cytoplasmic fraction. To resuspend the nuclei pellet, 500 µl of triton buffer (10 mM Tris, 100 mM NaCl, 1 mM EGTA, 300 mM sucrose, 0.5 mM $NaVO_3$, 50 mM NaF, 1 mM phenylmethylsulphonyl fluoride, 0.5% triton X-100, protease inhibitor cocktail, and RNasin). RNA was then extracted using Trizol.

### RNA FISH
RNA FISH was performed according to Marchese et al. (2016). Briefly, cells were fixed with 3.7% formaldehyde for 15 min. Fluorescein-labeled locked nucleic acid DNA probes (*ALAL-1* #1, 5′-ATATACCTGAGGTCTGCCAGGA-3′; #2 5′-ATCTGGGTCACC GAAACTGTA-3′) were synthesized by Exiqon and hybridized according to the manufacturer's protocol with some modifications. The probe-target RNA hybridization was performed overnight at 55°C. Probes' residues were eliminated through extensive washes with 2× SSC buffer, and fixed cells were incubated with 3% hydrogen peroxide for 30 min. For fluorescein (FAM) detection, cells were first incubated with blocking buffer (10% heat-inactivated goat serum, 0.5% Blocking Reagent [Roche; 11096176001] in PBS–0.5% Tween-20) and then with 1.5 U/ml of specific anti–FAM-POD antibody (Roche; 11426346910) diluted in blocking buffer. After washing, the signal was developed through incubation with TSA-Cy3 solution (Perkin Elmer). Slides were prepared for microscope imaging using mounting solution with DAPI. Fluorescent foci were quantified by imaging and counting ~100 cells per condition.

### Microscopy
Confocal images were acquired with a Zeiss Axio Imager M1 microscope equipped with a Plan-Apochromat 40×/0.95 Korr M27 objective lens or with a Zeiss Axio Observer Z1/7 with a Plan-Apochromat 63×/1.40 Oil DIC M27 objective lens. Images were captured with a Zeiss AxioCam MR R3 camera controlled with ZEN Software (Carl Zeiss). Acquisition information was as follows: EGFP (excitation 488, emission 509, detector: GaASP-Pmt1); Cy3 (excitation 548, emission 561, detector: GaASP-Pmt2); and DAPI (excitation 353, emission 465, detector: GaASP-Pmt3). Contrast adjustment and cropping were done with ImageJ or GIMP.

### Peripheral blood mononuclear cell (PBMC) isolation
PBMCs were isolated from healthy volunteers by density gradient centrifugation (1.077 *g* ml−1; Ficoll-Paque Plus; GE Healthcare). Cells were resuspended in RPMI 1640 culture medium supplemented with 10% (vol/vol) heat-inactivated, fetal bovine serum, penicillin–streptomycin (1%), and β-mercaptoethanol (50 µM; Sigma).

### Cell migration assays
In vitro cell migration assays were performed using 24-well Trans-well chambers (5-µm pore size; Costar). $3 × 10^5$ of PBMCs was added to the upper chamber and incubated for 5 h with A549 cells overexpressing *ALAL-1* or control cells in the

bottom well of the chamber. To determine the number of migratory cells, the lower cells were analyzed by flow cytometry with Perfect count microspheres (Cytognos) and fluorochrome-conjugated mAbs against CD3-BV421 (UCHT1), CD8-PeCy7 (RPA-T8), CD4-FITC (OKT4), CD19-PE (HIB19), CD56-APC (HCD56), and CD14-BV510 (M5E2). Samples were acquired on a FACSCanto-II cytometer (BD Biosciences). Data were analyzed using FlowJo software (TreeStar).

### Statistical analyses

Statistical comparisons were performed using Prism GraphPad and R (>3.01). Comparisons of *ALAL-1* expression and methylation in TCGA data were performed using a two-tailed unpaired *t* test with Welch's correction. Significance for tumor volumes in xenograft models was determined by a two-tailed Mann-Whitney test. Significance of *ALAL-1* expression in different immune phenotypes was determined by a two-tailed unpaired Wilcoxon test. All other significances were determined by a two-tailed unpaired *t* test. Data distribution was assumed to be normal, but this was not formally tested. Significances are represented as *, P ≤ 0.05; *, P ≤ 0.01; and *, P ≤ 0.001.

### Online supplemental material

Fig. S1 shows cancer-associated somatic copy number alterations (SCNAs) analysis. Fig. S2 shows that ALAL-1 promotes the oncogenic phenotype of different lung cancer cells. Fig. S3 shows that ALAL-1 is transcriptionally regulated by NF-kB. Fig. S4 shows that cancer phenotypes upon ALAL-1 depletion are not driven by *IKBKB*. Fig. S5 shows that ALAL-1 acts with SART3. Table S1 contains unique copy number–altered focal regions identified on SCNAs. Table S2 shows the effects of depletion of ALAL-1 and SART3 on gene expression in untreated and TNF-treated cells. Table S3 lists sgRNA sequences, genotyping, siRNA sequences, and primers used.

## Acknowledgments

We thank David Tamborero for his valuable advice and TCGA, ENCODE projects, and GEO for providing their platforms and the contributors for their valuable datasets.

A. Athie was supported by the Spanish Ministry of Economy Fellowship (BES-2012-055697). The research is supported by the European Research Council Starting and Consolidator grants 281877 and 771425 and by the Spanish Ministry of Economy grant BFU2014-58027-R.

The authors declare no competing financial interests.

Author contributions: A. Athie and M. Huarte designed the research; A. Athie, F.P. Marchese, J. González, T. Lozano, I. Raimondi, P.K. Juvvuna, A. Abad, O. Marin-Bejar, D. Ajona, and M.J. Pajares performed the research; A. Athie., M. Huarte, I. Raimondi, D. Martínez, J. Serizay, T. Lozano, C. Kanduri, and J.J. Lasarte performed data analysis/interpretation; and L.M. Montuenga and J. Sandoval provided and analyzed samples. M. Huarte and F.P. Marchese wrote the paper.

Submitted: 8 August 2019

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

# Supplemental material

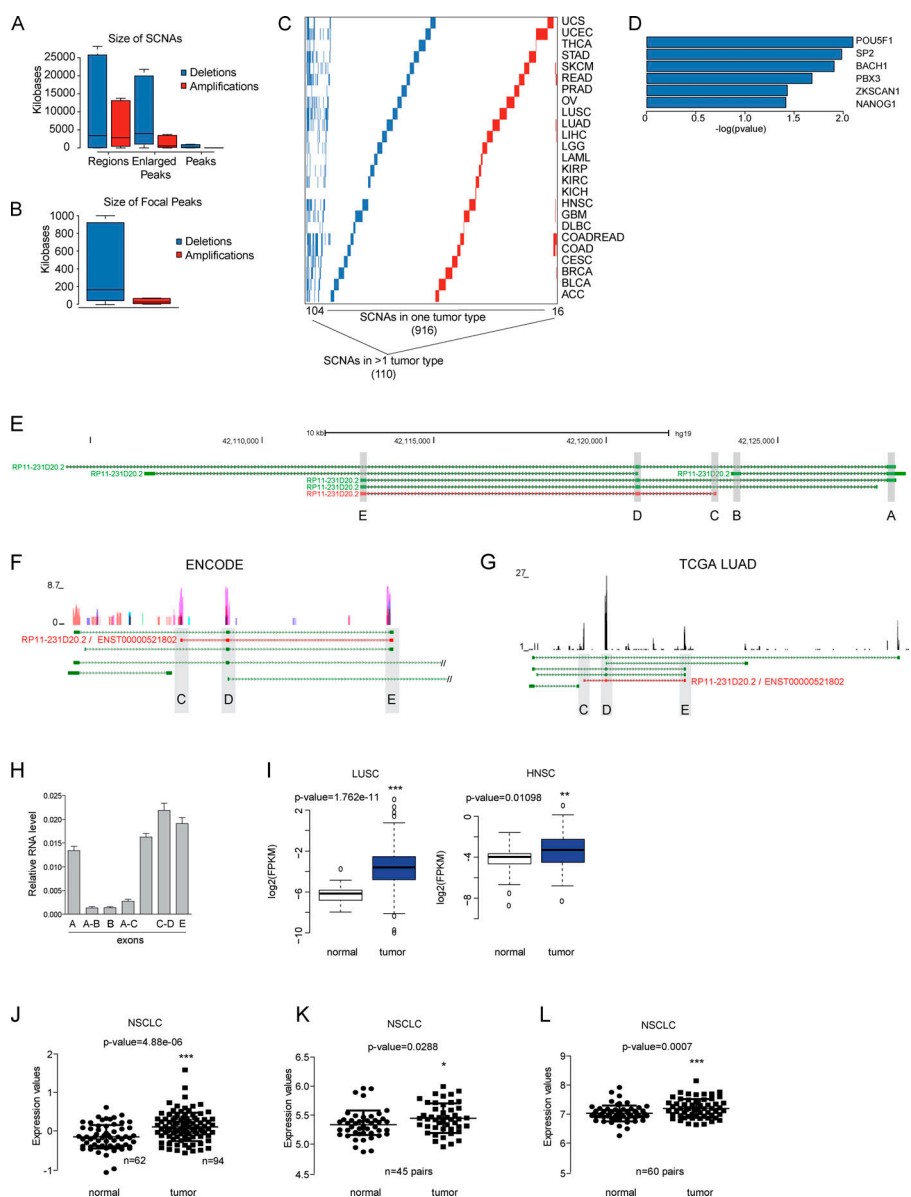

Figure S1. **Cancer-associated somatic copy number alterations (SCNAs) analysis. (A)** Size ranges of genomic deletions and amplifications at three levels: regions, enlarged peaks, and peaks. **(B)** Size range of the focal peaks reported by the GISTIC algorithm, defined as the part of the copy number alteration with the greatest amplitude and frequency. **(C)** Recurrence of the SCNAs identified, where rows represent SCNAs and columns represent tumor types. **(D)** Transcription factors with binding significantly enriched around the TSS of deleted lncRNAs. **(A–C)** Color code: deletions are shown in blue and amplifications in red. **(E)** Representation of the six different isoforms annotated in the *ALAL-1* locus with their Ensembl Transcript IDs. Isoforms are represented in the 5′ to 3′ direction, and exons are shadowed in gray and identified with letters (A–E). The ALAL-1 predominant form (RP11-231D20.2) is highlighted in red. **(F)** RNA-seq track showing the expression of *ALAL-1* in nine cell lines from the ENCODE project. **(G)** RNA-seq data from the TCGA-LUAD sample TCGA-44-7661-01A with a mean expression of *ALAL-1*, supporting the expression of ENST00000521802 (shown in red). **(H)** Relative expression of *ALAL-1* quantified by qRT-PCR with several sets of primers mapping to different exons. Error bars represent SEM. **(I)** Expression of *ALAL-1* in LUSC ($n$ = 398, T; $n$ = 44, N) and in head and neck squamous carcinoma (HNSC, $n$ = 297, T; $n$ = 37, N) tumors and normal samples. **(J–L)** *ALAL-1* expression quantified by microarray analysis (probe 231378_at) of cohort GSE19188, including 62 adjacent normal lung tissues and 94 NSCLC tumors (G); cohort GSE18842 with 45 paired samples (normal/tumor) of NSCLC (H); and cohort GSE19804, including 60 paired samples (normal/tumor) from nonsmoking female cancer patients (I–L). Statistical significance was determined by two-tailed unpaired $t$ test with Welch's correction and represented as *, $P \le 0.05$; **, $P \le 0.01$; ***, $P \le 0.001$. In A, B, and I, bottom and top of the box are the 25th and 75th percentile (the lower Q1 and upper quartile Q3), and the band near the middle of the box corresponds to the median. The lower whisker extends Q1 – 1.5 * interquartile range (IQR) and the upper one Q3 + 1.5 IQR. In F and G, green track represents the different isoforms of ALAL-1 annotated in Gencode v19. In J–L, the lines represent the median and the 25th and 75th percentile (the lower Q1 and upper quartile Q3). UCS, uterine carcinosarcoma; UCEC, uterine corpus endometrial carcinoma; THCA, thyroid carcinoma; STAD, stomach adenocarcinoma; SKCM, skin cutaneous melanoma; READ, rectum adenocarcinoma; PRAD, prostate adenocarcinoma; OV, ovarian serous cystadenocarcinoma; LIHC, liver hepatocellular carcinoma; LGG, brain lower grade glioma; LAML, acute myeloid leukemia; KIRP, kidney renal papillary cell carcinoma; KIRC, kidney renal clear cell carcinoma; GBM, glioblastoma multiforme; DLBC, lymphoid neoplasm diffuse large B-cell lymphoma; COADREAD, colon adenocarcinoma; COAD, colon adenocarcinoma; CESC, cervical squamous cell carcinoma and endocervical adenocarcinoma; BRCA, breast invasive carcinoma; BLCA, bladder urothelial carcinoma; ACC, adrenocortical carcinoma; T, tumor; N, normal; FPKM, fragments per kilobase of transcript per million mapped reads.

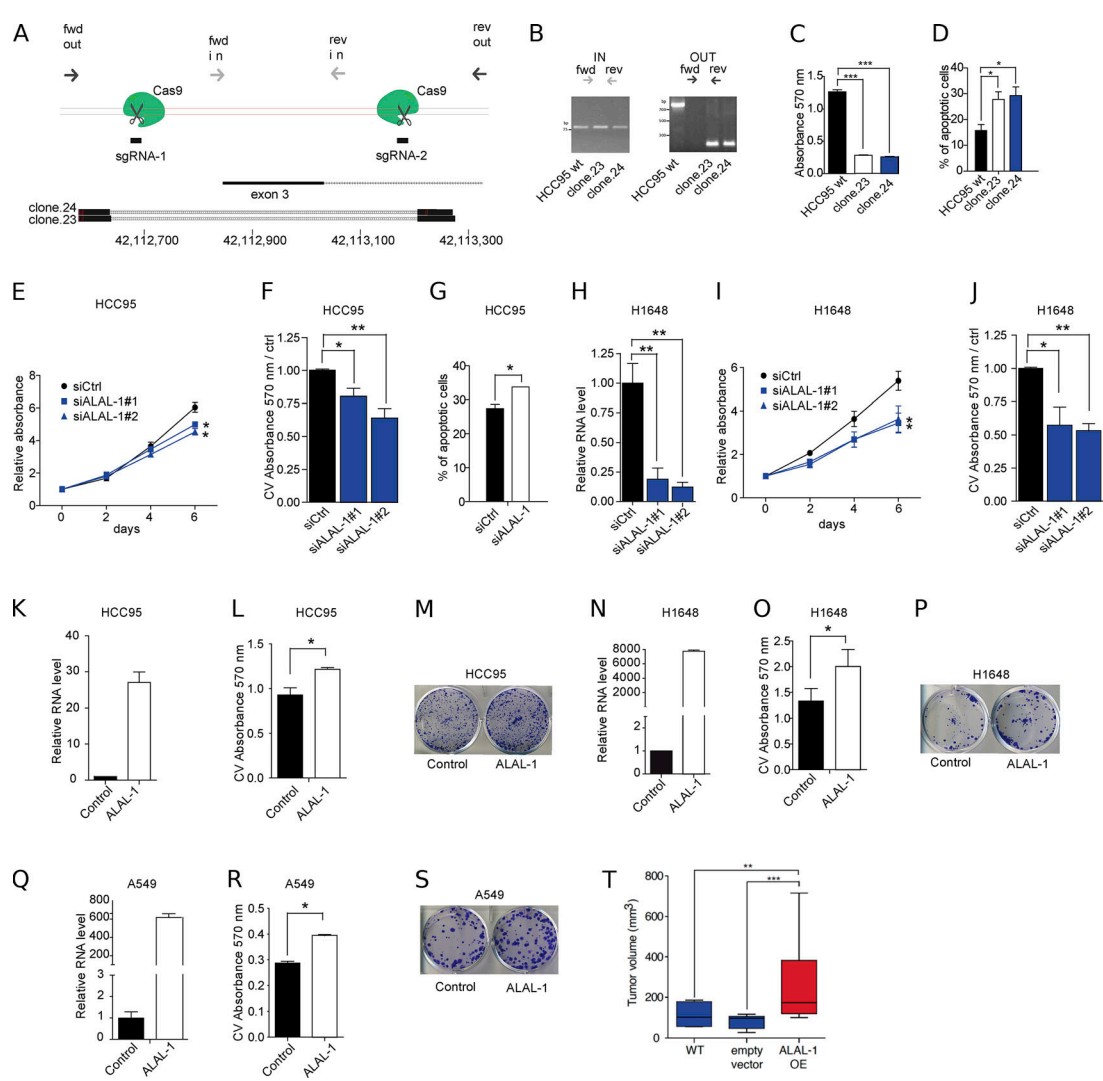

Figure S2.  **ALAL-1 promotes the oncogenic phenotype of different lung cancer cells. (A)** Diagram showing the location of the sgRNAs used for deleting a region of 500 bp flanking exon 3 of *ALAL-1*. The annealing sites for the primers used (forward [fwd]; reverse [rev]) to screen for the deletion are indicated. BLAT results of the obtained sequences of the PCR products show the flanking sequences of the deletion. **(B)** DNA electrophoresis showing the PCR products obtained with the fwd/rev in and out primers using genomic DNA from the parental HCC95 cells and clones (23 and 24). **(C)** Crystal violet absorbance measuring the colony formation capacity of the CRISPR/Cas9 engineered cells. **(D)** Percentage of apoptotic cells from the CRISPR/Cas9 clones in which *ALAL-1* expression was reduced. **(E)** Cell proliferation measured by MTS of HCC95 cells after *ALAL-1* inhibition. **(F)** Crystal violet (CV) absorbance measuring the colony formation capacity in HCC95 cells transfected with siRNAs (1 and 2) targeting *ALAL-1* or control siRNA. **(G)** Percentage of HCC95 apoptotic cells measured by flow cytometry after ALAL-1 inhibition. **(H)** Inhibition levels of *ALAL-1* obtained with siRNAs 1 and 2 in H1648 cells. RNA levels are normalized to HPRT and relative to the control siRNA. **(I)** Cell proliferation measured by MTS of H1648 cells after *ALAL-1* inhibition. **(J)** Crystal violet absorbance measuring the colony formation capacity in H1648 cells transfected with siRNAs (1 and 2) targeting *ALAL-1* or control siRNA. Absorbance values are represented relative to the control. **(K)** *ALAL-1* levels in HCC95 cells transiently transfected with the pcDNA3–*ALAL-1* plasmid. **(L and M)** Clonogenic assay of HCC95 cells transiently overexpressing *ALAL-1*. **(N)** *ALAL-1* levels in H1648 cells transfected with the pcDNA3–*ALAL-1* plasmid. **(O and P)** Clonogenic assay of H1648 cells transiently overexpressing *ALAL-1*. **(Q)** Overexpression levels of *ALAL-1* in A549 cells transduced with a retrovirus expressing the lncRNA. **(R and S)** Colony formation assay of A549 cells overexpressing *ALAL-1*. **(T)** Volumes of tumors formed by subcutaneous injection of wt, stably expressing empty vector, or ALAL-1–overexpressing A549 cells in immunocompromised mice. Boxplot of $n = 7$; significance was determined by one-tailed Mann-Whitney test. All other significances were determined by two-tailed unpaired *t* test and represented as *, P ≤ 0.05; **, P ≤ 0.01; ***, P ≤ 0.001. ctrl, control; siALAL-1, ALAL-1 siRNA; siCtrl, control siRNA. Error bars represent SEM. In T, bottom and top of the box are the 25th and 75th percentile (the lower Q1 and upper quartiles, respectively Q3), and the band near the middle of the box corresponds to the median. The lower whisker extends Q1 – 1.5 * interquartile range (IQR) and the upper one Q3 + 1.5 IQR.

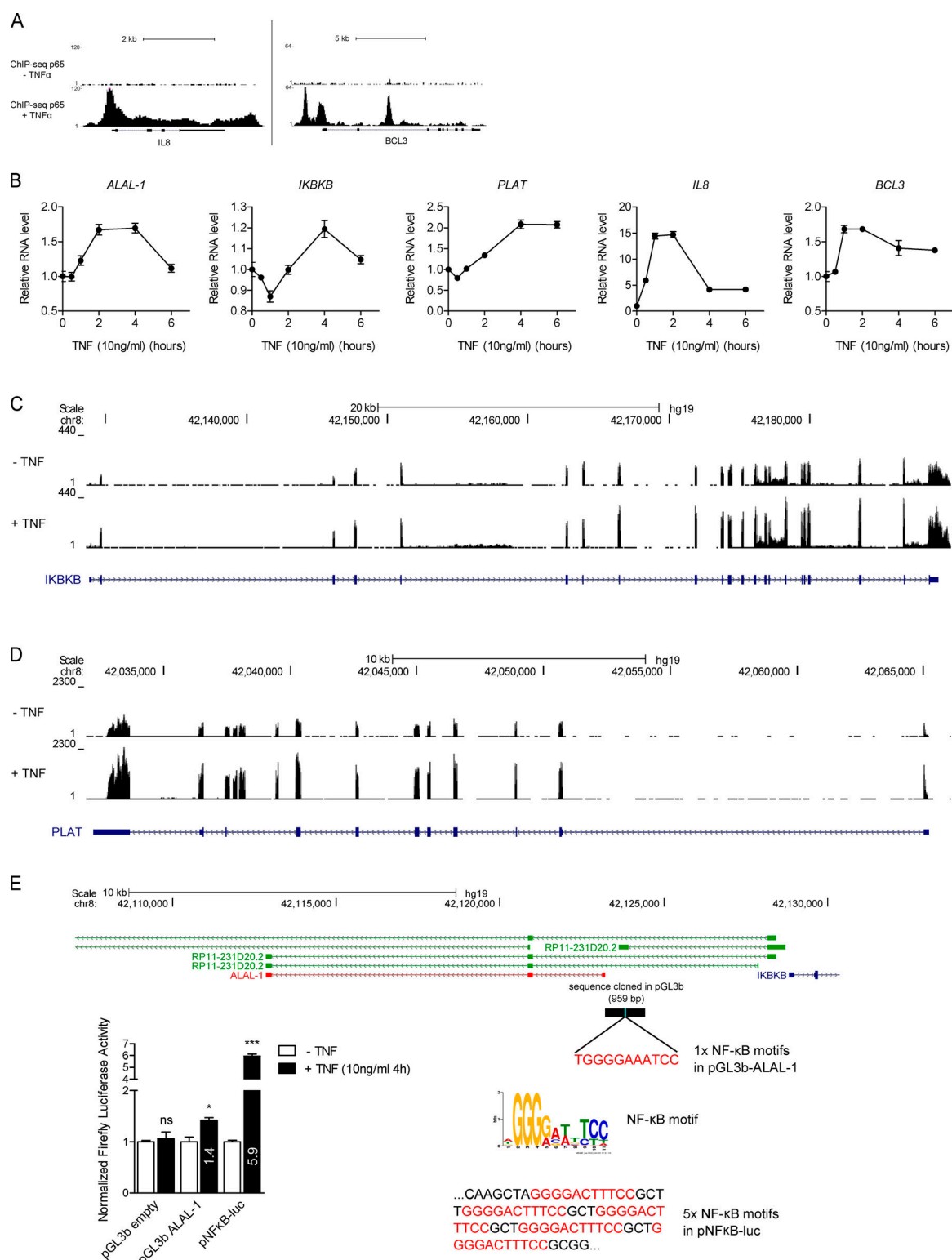

Figure S3.   **ALAL-1 is transcriptionally regulated by NF-kB. (A)** ChIP-seq signal of p65 protein at its known target loci *IL8* and *BCL3* in HUVEC cells untreated or treated with TNFα. **(B)** Time course experiment showing the induction by TNFα of *ALAL-1* and its neighbor genes, *IKBKB* and *PLAT*, and the p65 direct targets *IL8* and *BCL3* in HCC95 cells. **(C and D)** RNA-seq signal for *IKBKB* (C) and *PLAT* (D) in HCC95 treated or not treated with TNFα. **(E)** Schematic showing the *ALAL-1* genomic region (959 bp) containing one NF-κB motif that was cloned in pGL3b and assayed for luciferase reporter activity following TNFα treatment. Significance was determined by two-tailed unpaired *t* test and represented as *, P ≤ 0.05; ***, P ≤ 0.001. ns, not significant; HUVEC, human umbilical vein endothelial cells; p, plasmid. Error bars represent SEM.

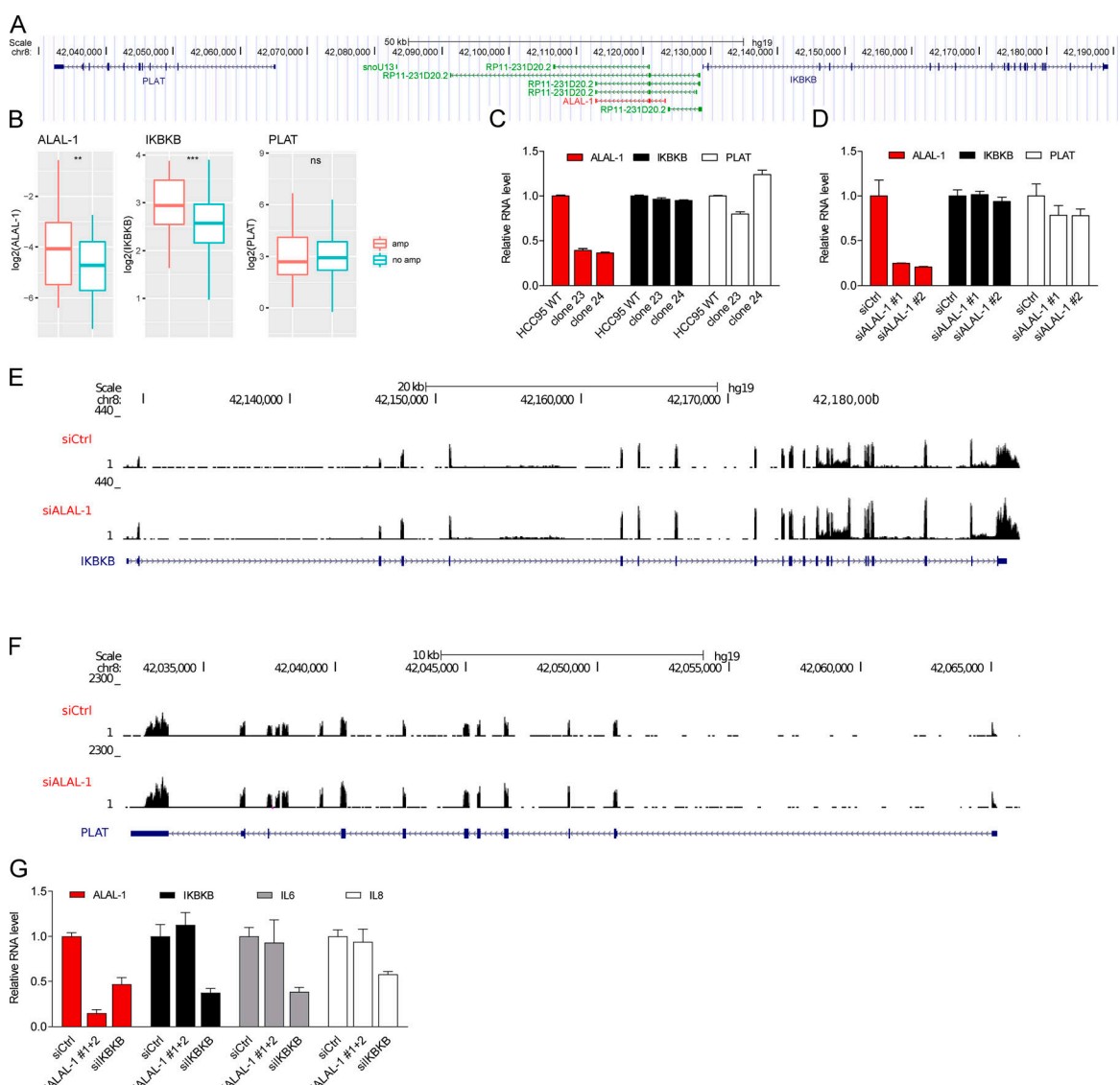

Figure S4. **Cancer phenotypes upon ALAL-1 depletion are not driven by *IKBKB*. (A)** View of the genomic region including *ALAL-1* and its neighbor genes, *IKBKB* and *PLAT*. **(B)** *ALAL-1*, *IKBKB*, and *PLAT* RNA levels in TCGA LUAD according to *ALAL-1* amplification status (amp, *n* = 43; no amp, *n* = 322). Statistical significance was determined by two-tailed unpaired *t* test with Welch's correction and represented as **, P ≤ 0.01; ***, P ≤ 0.001. Bottom and top of the box are the 25th and 75th percentile (the lower Q1 and upper quartiles, respectively Q3), and the band near the middle of the box corresponds to the median. The lower whisker extends Q1 – 1.5 * interquartile range (IQR) and the upper one Q3 + 1.5 IQR. **(C and D)** *ALAL-1*, *IKBKB*, and *PLAT* RNA levels as determined by qRT-PCR in HCC95 cells depleted of ALAL-1 by CRISPR (C) or by RNAi (D). **(E and F)** RNA-seq signal for *IKBKB* (E) and *PLAT* (F) in HCC95 depleted or not depleted of ALAL-1 by RNAi. **(G)** *ALAL-1, IKBKB, IL6,* and *IL8* RNA levels quantified by qRT-PCR in HCC95 depleted of ALAL-1 or IKBKB by RNAi. ns, not significant; siALAL-1, ALAL-1 siRNA; siCtrl, control siRNA; amp, amplification; siIKBKB, IKBKB-targeting siRNA. Graphs of mean ± SEM for three independent experiments are shown.

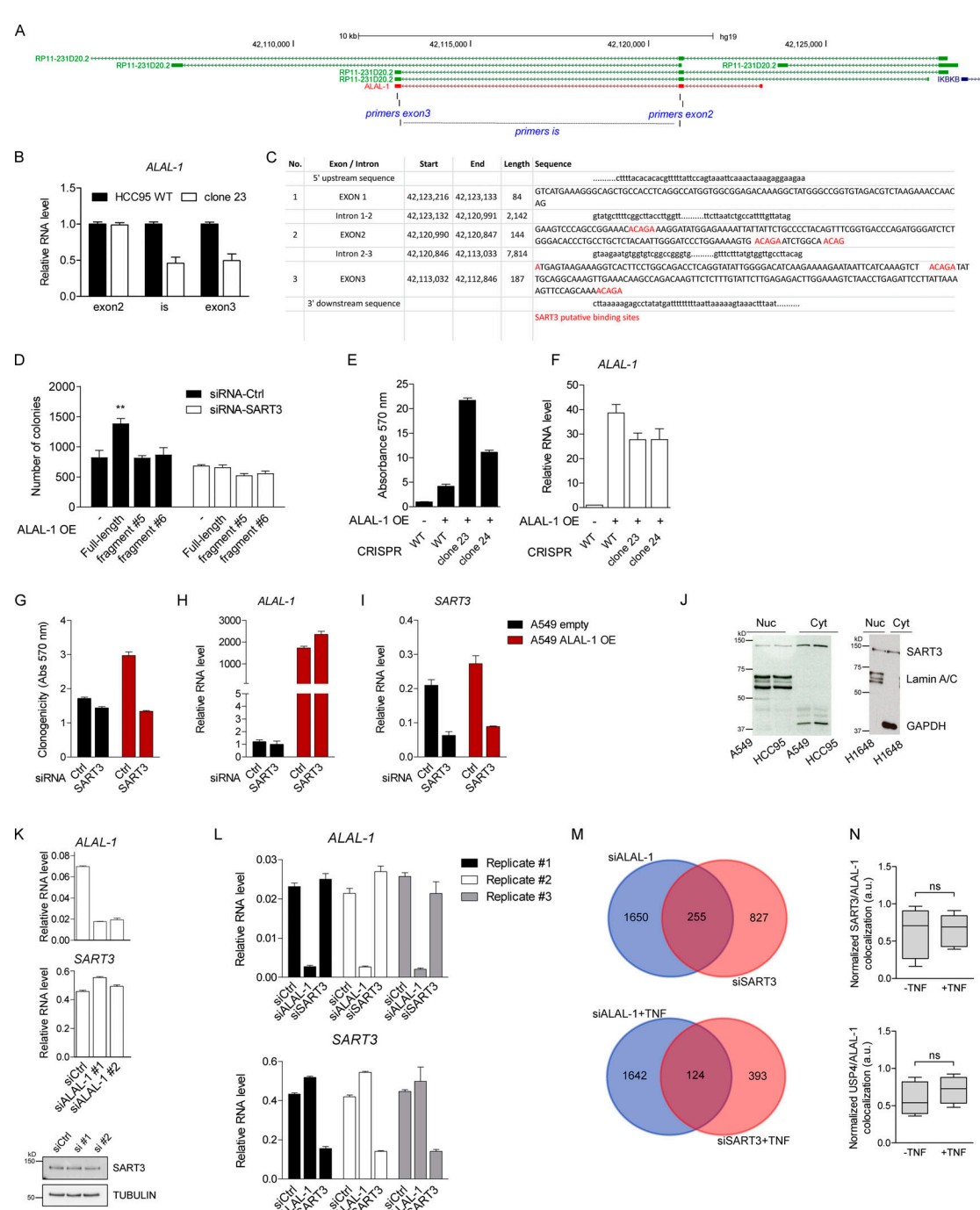

Figure S5. **ALAL-1 acts with SART3. (A)** View of the genomic region of *ALAL-1* with mapping of the qRT-PCR primers used to detect ALAL-1. **(B)** ALAL-1 levels in HCC95 WT and *ALAL-1* CRISPR depleted cells (clone 23) detected using different primer sets, as shown in A. **(C)** *ALAL-1* exon sequences (exons 1–3) with SART3 putative binding sites marked in red. **(D)** Clonogenicity assay of HCC95 cells transfected with *ALAL-1* (full length and fragments described in Fig. 5 D) and depleted or not depleted of SART3 by RNAi. **(E)** Clonogenicity assay of HCC95 cells depleted or not depleted of ALAL-1 by CRISPR and transfected with an overexpressing ALAL-1 vector. **(F)** ALAL-1 levels in cells used in D. **(G)** Clonogenicity assay of A549 cells stably overexpressing ALAL-1 or not (empty) and depleted of SART3 by RNAi or not (siRNA-Ctrl). **(H and I)** ALAL-1 and SART3 RNA levels in cells used in F. **(J)** SART3 localization in A549, HCC95, and H1648 cells identified by Western blot (anti–SART3 ab). Lamin A/C and GAPDH were used as controls of nuclear and cytoplasmic fractions, respectively. **(K)** Effect of ALAL-1 inhibition on SART3 mRNA and protein levels. **(L)** *ALAL-1* and SART3 mRNA levels quantified by qRT-PCR in the samples used for RNA-seq analyses. **(M)** Number of differentially expressed genes (adj. $P < 0.05$) determined by the RNA-seq analysis in the indicated conditions. **(N)** Graphs of colocalization quantifications of SART3/ALAL-1 and USP4/ALAL-1 (shown in Fig. 6) normalized to ALAL-1 signal. Graphs of mean (± SEM) for three independent experiments are shown. Significance was determined by two-tailed unpaired *t* test and represented as **, $P \leq 0.01$. Ctrl, control; ns, not significant; siALAL-1, ALAL-1 siRNA; siCtrl, control siRNA; OE, overexpression; Abs, absorbance; ab, antibody; Nuc, nucleus; Cyt, cytoplasm; siSART3, SART3-targeting siRNA. Bottom and top of the box are the 25th and 75th percentile (the lower Q1 and upper quartiles, respectively Q3), and the band near the middle of the box corresponds to the median. The lower whisker extends Q1 – 1.5 * interquartile range (IQR) and the upper one Q3 + 1.5 IQR.

Provided online are three Excel tables. Table S1 contains unique copy number–altered focal regions identified on SCNAs. Table S2 shows the effects of depletion of ALAL-1 and SART3 on gene expression in untreated and TNF-treated cells. Table S3 lists sgRNA sequences, genotyping, siRNA sequences, and primers used.

