## [Peer Review File · The Journal of Cell Biology]

Copy number alterations from a lncRNA perspective reveals a regulator of lung cancer immune evasion

Alejandro Athie, Francesco Marchese, Jovanna Gonzalez, Teresa Lozano, Ivan Raimondi, Prasanna Juvvuna, Amaya Abad, Oskar Marin-Bejar, Jacques Serizay, Dannys Martinez, Daniel Ajona, Maria Pajares, Juan Sandoval, Luis Montuenga, Chandrasekhar Kanduri, Juan Lasarte, and Maite Huarte

Corresponding Author(s): Maite Huarte, CIMA

Review Timeline:	Submission Date:	2019-08-08
	Editorial Decision:	2019-09-13
	Revision Received:	2019-12-12
	Editorial Decision:	2020-02-24
	Revision Received:	2020-05-08
	Editorial Decision:	2020-05-20
	Revision Received:	2020-05-25

Monitoring Editor: Ana Pombo

Scientific Editor: Tim Spencer

Transaction Report:

DOI: <https://doi.org/10.1083/jcb.201908078>

September 13, 2019

Re: JCB manuscript #201908078

Dr. Maite Huarte
CIMA
Pio XII, 55
Pamplona, 31008
Spain

Dear Dr. Huarte,

Thank you for submitting your manuscript entitled "The analysis of copy number alterations from a lncRNA perspective reveals a regulator of lung cancer immune evasion". Your manuscript has been assessed by expert reviewers, whose comments are appended below. We sincerely apologize for the delay in communicating our decision to you and thank you for your patience with the editorial and peer-review process. Although the reviewers express potential interest in this work, significant concerns unfortunately preclude publication of the current version of the manuscript in JCB.

You will see that the reviewers found the results interesting and the work largely of quality, but were critical of the depth of the cell biological analyses. Revs#1 and #2 suggested more work into the link between the lncRNA studied and the neighboring gene, IKBKB (Rev#1 #1, Rev#2 #1, #4, #6). Revs#2 and #3 suggested deeper investigations of the relationship between ALAL-1 and SART3 (Rev#2 points #5, #7; Rev#3 points #4 with various suggestions). Rev#3 suggested testing whether NFkB controls ALAL-1 transcription (#3). Rev#2 also asked for single-molecule resolution analyses of ALAL-1 (point #3). Rev#3 flagged overstated conclusions that could be toned down (point #2) and suggested validating the claim that ALAL-1 is an oncogene in vivo (#6). Rev#2 suggested removing the last figure (immuno studies).

Despite some of the reservations from the reviewers about the scope of the analyses, we appreciate the quality of the work and find that it provides an interesting advance in our understanding of tumor cell biology. We would be interested in considering a revision if you can further the cell biological analyses -- primarily to address the link between ALAL-1 and SART3 and clarify the relationship between ALAL-1 and IKBKB, since it could contribute to the phenotypes studied. Therefore, we feel that the goal in revision should be to clarify the bioinformatic analyses (e.g., respond to Rev#1 point #3 and Rev3's first two points in the text), deepen the mechanistic studies of the link between SART3 and ALAL-1, which have the potential for the greatest cell biological novelty to understand the basis for cancer cell behavior -- and Rev#3 has constructive, precise experimental suggestions -- if possible testing a direct transcriptional regulation of ALAL-1 by NFkB through luciferase assays, and clarify the impact of ALAL-1 expression on IKBKB. Examining the (in)dependency of IKBKB is especially important.

On the other hand, for publication in JCB, we would not require in vivo experiments (Rev#3) or single-molecule detection (Rev#2) given the scope of the paper and its strengths for a cell biological audience. We additionally recommend that you keep Figure 6, but consider adding any additional work that you may have available since submission to strengthen this part of the work (if already done). If this is not possible, toning down the conclusions related to Figure 6 would be

absolutely acceptable to us editorially.

Please let us know if you are able to address the major issues outlined above and wish to submit a revised manuscript to JCB. Note that a substantial amount of additional experimental data likely would be needed to satisfactorily address the concerns of the reviewers. Our typical timeframe for revisions is three to four months; if submitted within this timeframe, novelty will not be reassessed. We would be open to resubmission at a later date; however, please note that priority and novelty would be reassessed.

If you choose to revise and resubmit your manuscript, please also attend to the following editorial points. Please direct any editorial questions to the journal office.

GENERAL GUIDELINES:

Text limits: Character count is < 40,000, not including spaces. Count includes title page, abstract, introduction, results, discussion, acknowledgments, and figure legends. Count does not include materials and methods, references, tables, or supplemental legends.

Figures: Your manuscript may have up to 10 main text figures. To avoid delays in production, figures must be prepared according to the policies outlined in our Instructions to Authors, under Data Presentation, <http://jcb.rupress.org/site/misc/ifora.xhtml>. All figures in accepted manuscripts will be screened prior to publication.

IMPORTANT: It is JCB policy that if requested, original data images must be made available. Failure to provide original images upon request will result in unavoidable delays in publication. Please ensure that you have access to all original microscopy and blot data images before submitting your revision.

Supplemental information: There are strict limits on the allowable amount of supplemental data. Your manuscript may have up to 5 supplemental figures. Up to 10 supplemental videos or flash animations are allowed. A summary of all supplemental material should appear at the end of the Materials and methods section.

If you choose to resubmit, please include a cover letter addressing the reviewers' comments point by point. Please also highlight all changes in the text of the manuscript.

Regardless of how you choose to proceed, we hope that the comments below will prove constructive as your work progresses. We would be happy to discuss them further once you've had a chance to consider the points raised. You can contact the journal office with any questions, cellbio@rockefeller.edu or call (212) 327-8588.

Thank you for thinking of JCB as an appropriate place to publish your work.

Sincerely,

Ana Pombo, PhD
Editor, Journal of Cell Biology

Melina Casadio, PhD
Senior Scientific Editor, Journal of Cell Biology

Reviewer #1 (Comments to the Authors (Required)):

The manuscript entitled "The analysis of copy number alterations from a lncRNA perspective reveals a regulator of lung cancer immune evasion" is well-written and well-structured. It presents an interesting, timely and original study identifying a novel non-coding transcript functioning in cancer. The experiments are very well-controlled and support the conclusions drawn and the model proposed. Prior to publication, several - mostly minor - issues should be addressed:

1) Gene Locus: IKBKB

As also mentioned in the manuscript, the RP11-231D20.2 lncRNA (ALAL-1) shares the genomic locus with a neighboring protein coding gene, IKBKB, in head-to-head antisense orientation. Given the obvious link of IKBKB to the TNF and immune pathways linked to ALAL-1 and given the finding, that this shared promoter contains p65/RelA binding sites further linking it to TNF and NF- κ B signaling, I would like to suggest further investigating - or excluding - a potential contribution of IKBKB. As a first step, the expression and correlation to ALAL-1 of IKBKB should be analyzed in the whole-transcriptome datasets used in the different parts of the study as well as IKBKB should be tested by RT-qPCR in the CRISPR-mediated deletion clones for ALAL-1. Depending on the results, additional experiments dissecting ALAL-1 and IKBKB function may be desirable.

2) Cell Lines

Cell line unique identifiers should be provided in the Methods section and the full names of the cell lines should be used (e.g. NCI-H1648). Also, information about cell line authentication and routine mycoplasma testing should be mentioned.

3) RNA Copy Numbers

More details on the approach taken for estimation of RNA copy numbers per cell (estimated with 150 by RT-qPCR and FISH) should be provided in the Methods section.

Reviewer #2 (Comments to the Authors (Required)):

The manuscript by Athie et al surveys somatic copy number alterations across multiple human cancer types and identifies candidate regions associated with both known and uncharacterized lncRNAs. In particular, they focus on the lncRNA they name ALAL-1, which shows evidence for amplification and overexpression in different subtypes of human cancer. For loss-of-function studies, they perform siRNA-mediated knockdown as well as CRISPR-mediated deletion of ~500 bp region that includes exon 3 of ALAL-1. They observe that these perturbations of ALAL-1 lead to reduced proliferation and colony formation capacity. For gain-of-function studies, they perform exogenous overexpression of ALAL-1 and note increased clonogenicity. Mechanistically, the authors propose a model where cytoplasmic ALAL-1 interacts with the splicing factor SART3 through a central region. In their model, ALAL-1 modulates noncanonical roles of SART3 in the ubiquitin degradation pathway that could potentially affect broad networks such as the Wnt/b-cat, NF- κ B, p53 and TGF β pathways. Finally, the authors focus on a subset of these signaling pathway, which are involved in the regulation of inflammatory processes and they identify a negative correlation between ALAL-1 levels and immune infiltration in vivo and cytokine secretion in vitro.

This work has potential to make a valuable contribution to the field of lncRNAs but, in its current

form, lacks depth in multiple key areas. In particular, careful and expanded analysis of the RNA contribution (points 1-5) and omission of more translational aspects of the study (points 6-8) may make it a more suitable candidate for publication at JCB.

- 1) The authors need to provide further characterization of the CRISPR deletion and its effect on lncRNA production and neighboring gene expression. Is transcription, levels and stability of upstream exons affected? Do other splice isoforms become more abundant? What is the effect on the neighboring gene IKBKB, which is a component of one of the pathways they suggest ALAL-1 regulates and is regulated by? Can they show through epistasis experiments that ALAL-1 and IKBKB act independently?
- 2) The authors should provide a complete set of rescue experiments in RNAi and CRISPR deletion settings with both FL as well as truncation mutants (including mutant lacking SATR3 binding domain).
- 3) The authors should use smRNA-FISH to detect ALAL-1 at single molecule resolution.
- 4) The authors need to discuss in more detail the implications of ALAL-1 sharing promoter with IKBKB, for example for the interpretation of the methylation experiments and throughout the text. Do ALAL-1 and IKBKB show correlated expression in cancer samples?
- 5) The authors need to provide further experimental evidence that the growth and colony formation phenotypes are mediated through SART3. Beyond correlative comparisons of gene expression analysis it is not clear that ALAL-1 and SART3 act together as a complex.
- 6) Fig. 4 focuses on the transcriptional regulation of ALAL-1 by NF kB. The authors need to show that this is independent of IKBKB-associated promoter and potential enhancer regions.
- 7) The relationship between ALAL-1, SART3 and downstream pathways is largely correlative and needs further experimental support for functional interplay (Fig. 5).
- 8) The initial evidence suggest that ALAL-1 acts through cell autonomous mechanisms comes in conflict with the evidence for immune-modulatory phenotypes. Fig. 6 is weak and lacks depth, functional or mechanistic insights. The manuscript is perhaps better off without it.

Reviewer #3 (Comments to the Authors (Required)):

This is a study by Athie et al, describing identification and characterization of lncRNA, ALAL-1 that is focally amplified in lung cancers. The authors analyzed somatic copy number alterations (SCNAs) using publicly available databases, detected the numerous SCNAs, and nailed down to previously uncharacterized ALAL-1 lncRNA. ALAL-1 is one of the targets of NF-kB. ALAL-1 regulates USP4 subcellular localization via tumor-rejection antigen START3. Indeed, ALAL-1 positively regulates proliferation of non-small lung cancer cell lines. In ALAL-1 overexpressing lung cancer cell lines, the level of cytokines decreased, resulting in disabling a migration potency of several immune cells. With these observations, the authors suggest that ALAL-1 is a proto-oncogenic lncRNA that mediates cancer immune evasion.

General comments:

Overall, the authors provide a potentially interesting concept that lung cancer cells overexpressing lncRNA ALAL-1 can evade host immune system. Their bioinformatic strategy to find new and interesting lncRNAs among SCNAs may be powerful. However, this is not novel, because a similar work identifying lncRNA genes in focal SCNAs among several types of cancers has been reported elsewhere (PMID:25203321). A major concern is that the manuscript lacks convincing mechanistic insight into cellular function. The authors heavily depend on the bioinformatic results and previous knowledge, and does not fully and experimentally develop the main conclusion. Sometimes, the author's conclusion is over-stated. Besides, there are typos and mislabeling in the manuscript.

Specific comments (major):

1. Interpretation of Figure 2C-E are confusing. I understand that ALAL-1 is highly expressed in the amplified group (Figure 2C) and in tumor (Figure 2D). However, majority of tumors (~ 66%) highly express ALAL-1 without the amplification of the ALAL-1 locus (Figure 2E). Moreover, ALAL-1 was also overexpressed in other types of tumors, such as LUSC, where ALAL-1 was not identified as frequently amplified (Supplementary Figure S2E). Therefore, importance of the ALAL-1 amplification in its overexpression and tumors is not clear.

First, the authors should not provide wrong impression that ALAL-1 is frequently amplified in cancer, which leads to its overexpression. A population of lung cancer with high expression and amplification of ALAL-1 is a minor (~7.2%) (Fig. 2E). They should describe the reason why they focus on this particular population.

Second, the authors should make it clear whether amplification of ALAL-1 gene is a cause of the high expression of ALAL-1.

2. In the abstract, the authors claim that ALAL-1 is overexpressed through epigenetic mechanisms: "ALAL-1 is also overexpressed in additional tumor types, such as lung squamous carcinoma through epigenetic mechanisms (line 7, page 2)". This is overstating. Bioinformatics analysis detected the two differentially methylated CpG sites (Figure 2 F-H). These show a correlation, but not enough evidence for "epigenetic mechanisms".

3. If the authors conclude that ALAL-1 is a transcriptional target of NF- κ B, it should be tested by a promoter luciferase assay whether NF- κ B directly acts as a transcriptional factor towards ALAL-1 expression in cultured cell lines. In addition, it is interesting to know whether NF- κ B expression also correlate with levels of ALAL-1 expression in the patient samples.

4. The authors claim that the specific interaction between ALAL-1 and SART3 is crucial for the cellular localization of USP4, which potentially activate USP4's function on the NF- κ B pathway. This argument is a key mechanistic insight of this manuscript, and should be addressed in more detail, as follows:

(4-1)

Although the authors detected ALAL-1 predominantly in the cytoplasm, it is not still clear whether ALAL-1 works in the nucleus or cytoplasm. Does ALAL-1 and SART3 make a complex mainly in the cytoplasm? In Fig.5B, the authors should perform the RIP assay using the cell lysates fractionated into the nucleus and cytoplasm, respectively. Also, they should address whether the level of this RNA-protein complex increase upon TNF- α

(4-2)

Upon TNF- α -treatment, does the FISH signal of ALAL-1 colocalize with that of HA-START3 in the cytoplasm?

(4-3)

The authors found that the SART3-interaction region is located in the middle region of ALAL-1 (Fig. 5D). Accordingly, the authors should create the mutant ALAL-1 which does not associate with SART3. Then, (1) they should ask whether overexpression of this mutant version of ALAL-1 fails to rescue the cellular localization of USP4, using the assay shown in Fig. 5F-G. (2) They should ask whether this mutant ALAL-1 influence the expressions of cytokines. (3) They should also ask whether this mutant ALAL-1 influences the proliferation of lung cancer cell lines (Supplementary Fig.4).

(4-4)

The authors also should make the mutant version of SART3 which cannot associate with ALAL-1, and ask whether overexpression of this mutant SART3 affect the cellular localization of USP4, using the assay shown in Fig.5G.

(4-5)

Does depletion of ALAL-1 disrupt the SART3-USP4 complex?

(4-6)

Upon depletion of ALAL-1, the SART3-USP4 complex does not localize properly (Fig.5F). Does this mean that this complex would not exert its deubiquitination ability toward the substrates upon depletion of ALAL-1? If so, the authors should show one or more substrates (factors involved in the NF- κ B pathways) whose poly-(or mono-) ubiquitination decreased upon ALAL-1-depletion.

5. In Fig.5C, it lacks control experiment using RNase, which can show back ground levels of this experiment.

6. The authors claim that ALAL-1 is pro-oncogene. It will be nice if the authors can show its oncogenic effects of ALAL-1 in mice, by xenograft of ALAL-1 overexpressing cell lines.

Specific comments (minor):

1. The authors should carefully cite appropriate references in the text. On page 10, in the last paragraph, ref. #32, 33, 41, 42 and 43 are cited in wrong place.

2. On page 10, lines 3 and 5, wrong figure numbers are referred (Supplementary Fig. 5E, instead of 5C).

3. On page 10, lines 28 and 29, wrong figure numbers are referred (Figure 6A-B).

4. I could not understand why the agarose gel in Supplementary Fig. 3B (right, "OUT") does not show the parental slow-migrating band. If the deletion by CRISPR was heterozygous in clones 23 and 24, as the authors described in the fourth paragraph on page 7, the clones must contain both the deleted and intact alleles.

5. There are typos in the Figure legends, as follows.

- ...de lncRNA PR11..... (Figure 1 G, on page 19, line 16)

- CAN 623 should be read as CNA623 (Figure 2A, on page 19, line 22).

- "...in read." should be read as "...in red." (Figure 2A, on page 19, line 23) .
- "...exon D..." should be read as "...exon 3..."(Figure 3E, on page 20, line 14)
- The sentence "...HUVEC cells treated or and treated with..." should be corrected. (Figure 4A, on page 21, line 1)
- The sentence "ALAL-1 interacts with regulates key cellular key cellular..." should be corrected (Figure 5, on page 21, line 16)
- "E. Number of cells of..." is incorrectly labeled as "C. Number of cells of..." (Figure 6, on page 22, line 11).
- "...500 bp flanking exon D..." should be read as "...500 bp flanking exon 3..." (in the legend for Supplementary Figure 3A).

6. There are possible typos in the Method section, as follows.

- In the text and the Figure 6 legend, the authors describe that they used A549 cells for the cell migration and cytokine production assays (lines 31 and 33 on page 11, and lines 9 and 11 on page 22). Regardless, they describe that they used HCC95 cells, in the Method section. They have to correct the discrepancy, for the sake of potential readers.
- In the Immunofluorescence section, the antibodies detecting HA tag (or the endogenous SART3?) used for Figure 5F should be described.

Response to Reviewers

We are most grateful to the three reviewers for previously taking the time to read our manuscript and for providing very useful feedback. We very much appreciate that reviewers found the original paper interesting and potentially significant. We have been stimulated by the points raised by the reviewers to perform further experimental work, and trust that the reviewers will now agree that this has improved the paper sufficiently to render it acceptable for publication in Journal of Cell Biology.

We now include new data that strengthen our conclusion on ALAL-1 being regulated by NF- κ B (new Supplementary Fig.3), clarify the relationship between ALAL-1 and its neighbour gene IKBKB (new Supplementary Fig.4), as well as reinforce our model on the physical and functional interaction between ALAL-1 and SART3/USP4 (new Figure 6, new Supplementary Fig.2T, Supplementary Fig.5A-H and Supplementary Fig.5M).

Our responses to the individual points raised by the reviewers are presented on a point- by-point basis below.

Reviewer #1 (Comments to the Authors (Required)):

The manuscript entitled "The analysis of copy number alterations from a lncRNA perspective reveals a regulator of lung cancer immune evasion" is well-written and well-structured. It presents an interesting, timely and original study identifying a novel non-coding transcript functioning in cancer. The experiments are very well-controlled and support the conclusions drawn and the model proposed. Prior to publication, several - mostly minor - issues should be addressed:

We thank the reviewer for his/her overall positive assessment of our work and suggestions.

1) Gene Locus: *IKBKB*

As also mentioned in the manuscript, the RP11-231D20.2 lncRNA (*ALAL-1*) shares the genomic locus with a neighboring protein coding gene, *IKBKB*, in head-to-head antisense orientation. Given the obvious link of *IKBKB* to the TNF and immune pathways linked to *ALAL-1* and given the finding, that this shared promoter contains p65/RelA binding sites further linking it to TNF and NF- κ B signaling, I would like to suggest further investigating - or excluding - a potential contribution of *IKBKB*. As a first step, the expression and correlation to *ALAL-1* of *IKBKB* should be analyzed in the whole-transcriptome datasets used in the different parts of the study as well as *IKBKB* should be tested by RT-qPCR in the CRISPR-mediated deletion clones for *ALAL-1*. Depending on the results, additional experiments dissecting *ALAL-1* and *IKBKB* function may be desirable.

We agree with the reviewer, and indeed a possible effect on the neighbour gene *IKBKB* was our first hypothesis as a plausible role of *ALAL-1*. We performed a number of analyses to test this hypothesis. We now include the results obtained in Suppl. Fig.6. Analysis of the TCGA LUAD data, where *ALAL-1* is found amplified, indicate that the genomic locus including *ALAL-1* and its neighbour gene *IKBKB* shows a certain degree of co-amplification and increased expression in lung adenocarcinoma, which is expected due to their genomic proximity and the average size of the focal amplifications (Figure 2A and Suppl. Fig.4B). However, direct depletion of *ALAL-1* by CRISPR or RNAi shows no effect on the mRNA levels of *IKBKB* by RT-qPCR (Suppl. Fig.4C) and RNA-seq (Suppl. Fig.4E), indicating that the effects caused by *ALAL-1* depletion are independent of *IKBKB*, and, although both are co-regulated by p65/RelA, *ALAL-1* acts independently of *IKBKB*.

We propose that *ALAL-1* and *IKBKB* co-regulation is the result of the orchestrated activation of the multiple genes that contribute to the TNF-alpha cellular response. However *ALAL-1* and *IKBKB* are independent actors of such response.

Please, see also response to Reviewer #2 point 1).

2) Cell Lines

Cell line unique identifiers should be provided in the Methods section and the full names of the cell lines should be used (e.g. NCI-H1648). Also, information about cell line authentication and routine mycoplasma testing should be mentioned.

We have corrected it in the Methods section and included the following information:

“Short tandem repeat (STR) profiling was used to authenticate cell lines and cells were tested for mycoplasma contamination regularly using the MycoAlert Mycoplasma Detection Kit (Lonza)”

3) RNA Copy Numbers

More details on the approach taken for estimation of RNA copy numbers per cell (estimated with 150 by RT-qPCR and FISH) should be provided in the Methods section.

We have added more details in the Methods section:

“To evaluate the absolute number of *ALAL-1* RNA molecules per cell, total RNA was isolated from HCC95 cells accurately counted using a Countess Automated Cell Counter (Thermo Fisher). RNA extraction and cDNA generation was performed as described using 1µg of RNA. Standard curve was obtained by qPCR of serial dilutions of a known amount of *in vitro* transcribed *ALAL-1* transcribed and used to calculate the copy number of *ALAL-1* per cell.”

“Fluorescent foci were quantified by imaging and counting approximately 100 cells per condition”

Reviewer #2 (Comments to the Authors (Required)):

The manuscript by Athie et al surveys somatic copy number alterations across multiple human cancer types and identifies candidate regions associated with both known and uncharacterized lncRNAs. In particular, they focus on the lncRNA they name *ALAL-1*, which shows evidence for amplification and overexpression in different subtypes of human cancer. For loss-of-function studies, they perform siRNA-mediated knockdown as well as CRISPR-mediated deletion of ~500 bp region that includes exon 3 of *ALAL-1*. They observe that these perturbations of *ALAL-1* lead to reduced proliferation and colony formation capacity. For gain-of-function studies, they perform exogenous overexpression of *ALAL-1* and note increased clonogenicity. Mechanistically, the authors propose a model where cytoplasmic *ALAL-1* interacts with the splicing factor *SART3* through a central region. In their model, *ALAL-1* modulates noncanonical roles of *SART3* in the ubiquitin degradation pathway that could potentially affect broad networks such as the Wnt/b-cat, NF-Kb, p53 and TGFb pathways. Finally, the authors focus on a subset of these signaling pathway, which are involved in the regulation of inflammatory processes and they identify a negative correlation between *ALAL-1* levels and immune infiltration in vivo and cytokine secretion in vitro.

This work has potential to make a valuable contribution to the field of lncRNAs but, in its current form, lacks depth in multiple key areas. In particular, careful and expanded analysis of the RNA contribution (points 1-5) and omission of more translational aspects of the study (points 6-8) may make it a more suitable candidate for publication at JCB.

We thank the reviewer for considering our work a potential valuable contribution to the

field and appreciate his/her suggestions.

1) The authors need to provide further characterization of the CRISPR deletion and its effect on lncRNA production and neighboring gene expression. Is transcription, levels and stability of upstream exons affected? Do other splice isoforms become more abundant? What is the effect on the neighboring gene *IKBKB*, which is a component of one of the pathways they suggest *ALAL-1* regulates and is regulated by? Can they show through epistasis experiments that *ALAL-1* and *IKBKB* act independently?

We agree with the reviewer and now provide further characterization of the *ALAL-1* CRISPR deletion clones, suggesting that *ALAL-1* acts independently of *IKBKB* and that its function requires the sequence contained in the exon 3.

Our reasoning for deleting the exon 3 of *ALAL-1* comes from the fact that deletion of the first or second exon of *ALAL-1* - proximal to the 5' regulatory region shared with its neighbour gene *IKBKB* - could likely have had an effect on *IKBKB* expression not solely dependent on the RNA product of *ALAL-1*, impeding the assessment of *ALAL-1* exclusive function, the goal of our experiment.

With the new presented data we show that deletion of the exon 3 of *ALAL-1*, which contains more than half of the sequence of the lncRNA, does not affect the levels of *ALAL-1* neighbour genes *IKBKB* and *PLAT* (Suppl.Fig.4C-D), while clearly impairing the function of the lncRNA (Fig.3). Additionally, *ALAL-1* CRISPR depletion shows no changes in the RNA level of the lncRNA when qPCR primers mapping to *ALAL-1* exon 2 were used (Suppl.Fig.5A-B), suggesting that neither the transcription of other isoforms nor *ALAL-1* is affected by the CRISPR depletion of *ALAL-1* exon 3. Moreover, we mapped the putative binding sites of SART3 to *ALAL-1*, showing their location between the second and third exons of the lncRNA (Suppl.Fig.5C), which together with our SART3 binding mapping to *ALAL-1* (Fig.5D), suggest that the exon 3 sequence of the lncRNA is required for its function.

Please, see also answer to Reviewer #1 point 1.

2) The authors should provide a complete set of rescue experiments in RNAi and CRISPR deletion settings with both FL as well as truncation mutants (including mutant lacking SART3 binding domain).

We thank the reviewer for the suggestion, and now include in Suppl.Fig.5D-E additional results obtained by rescuing *ALAL-1* expression in both *ALAL-1* CRISPR clones used in the study. For this, *ALAL-1* was cloned into pcDNA3 and transfected in HCC95 cells, WT CRISPR control or *ALAL-1* CRISPR KO clones (clone 23 and clone 24), and clonogenicity assays were performed. *ALAL-1* over-expression (OE) in clone 23 and clone 24 strongly increased the number of cell colonies. To a minor extent, the number of colonies was also increased in HCC95 WT cells when *ALAL-1* was over-expressed as previously observed (Suppl. Fig.2K-M). This, together with our other data, indicates that *ALAL-1* RNA itself has a role in the oncogenic phenotype of lung

cancer cells.

To address the specific role of different fragments of *ALAL-1*, we cloned three of the *ALAL-1* fragments described in Fig.5D (fragments 4-6) into pcDNA3 and performed clonogenicity assays as described above using HCC95 clone 24 cells. However, while the full-length version of *ALAL-1* was able to enhance cell clonogenicity as previously observed, we did not observe any difference in colony formation when the truncated forms of *ALAL-1* were overexpressed (Reviewer Fig.1). This data indicates that *ALAL-1* truncations are not sufficient to induce a phenotype of increased cell proliferation, suggesting that the full sequence of the RNA is required to achieve its functions. It is plausible that other functional interactions are mediated through additional regions of the RNA and/or structural conformation of the truncated fragments is not adequate. Nevertheless, these *ALAL-1* fragments remain useful tools for future *in vitro* experiments to investigate protein binding or RNA structure.

Reviewer Fig.1

Full length *ALAL-1* and fragments (described in Fig.5D) were cloned into pcDNA3 and transfected in HCC95 clone 24 cells. The transfection of the empty plasmid was used as control. Clonogenicity assays were then performed.

3) The authors should use smRNA-FISH to detect *ALAL-1* at single molecule resolution.

We agree with the reviewer that smRNA-FISH allows a better resolution of the RNA molecules and indeed smRNA-FISH using Stellaris® RNA probes is generally our first choice approach in the laboratory and we have used it before in other studies (for instance, Marín-Béjar, O. et al., *Genome Biol.* 14:R104). We tried to apply to *ALAL-1* with no success. For some lncRNAs this approach is not feasible due to the shorter length of the RNA, which limits the design of the pool of FISH RNA probes. This is also the case of *ALAL-1*, which is 415 nt long and failed to pass the requirements set by the *Stellaris probe designer* (usually a minimal of 30 tiling probes of 20 nt long each). Therefore, we had to choose LNA probes as alternative RNA-FISH approach, which is suited for the efficient detection of noncoding RNAs.

4) The authors need to discuss in more detail the implications of *ALAL-1* sharing promoter with *IKBKB*, for example for the interpretation of the methylation experiments and throughout the text. Do *ALAL-1* and *IKBKB* show correlated expression in cancer samples?

As also suggested by Reviewer #1 and #3, we now include more data on the relationship between *ALAL-1* and its neighbour gene *IKBKB* in Suppl.Fig.4, as well as discuss it in the text. Moreover, as suggested, we analysed the expression of *ALAL-1* in relation to its neighbour genes *IKBKB* and *PLAT* in LUAD and LUSC TCGA datasets and observed a weak positive correlation between *ALAL-1* and either two proximal genes ($0.25 \leq R \leq 0.5$) (Reviewer Fig.2), suggesting that *ALAL-1* and *IKBKB* or *PLAT* are transcriptionally regulated in an independent or only partially dependent manner.

We speculate that the co-regulation by p65/RelA of *ALAL-1* and *IKBKB* allows the co-activation of two independent mechanisms in response to pro-inflammatory stimuli. Indeed co-activation by transcription factors is frequently occurring in cellular networks to ensure an efficient and coordinated response.

In summary, our data show that although *ALAL-1* and *IKBKB* share some regulatory elements, *ALAL-1* does not regulate the expression of *IKBKB*. On the contrary, it has an independent function within the inflammatory response.

Reviewer Fig.2

Correlation of *ALAL-1* expression with *IKBKB* and *PLAT* in TCGA LUAD and LUSC datasets.

5) The authors need to provide further experimental evidence that the growth and colony formation phenotypes are mediated through SART3. Beyond comparative gene expression analysis it is not clear that *ALAL-1* and SART3 act together as a complex.

We show that depletion of *ALAL-1* by CRISPR editing or RNAi reduces the proliferation and number of colonies of different cancer cells (Fig.3H-I, Suppl. Fig.2), while OE of *ALAL-1* favours colony formation (Suppl. Fig.2). To get more insight into the connection between *ALAL-1* and SART3 in terms of cellular phenotype we used

A549 cells, with low basal *ALAL-1* expression. As already observed before, OE of *ALAL-1* favours colony formation. In the absence of *ALAL-1* OE, SART3 knockdown only minimally reduces clonogenicity (Suppl. Fig.5F-H). However, when SART3 was depleted in *ALAL-1* OE cells, the acquired ability to form more colonies was lost (Suppl. Fig.5F-H). The results suggest that *ALAL-1* requires the presence of SART3 to promote clonogenicity and together with the other data presented in the manuscript support the model we propose that sees *ALAL-1* as a physical and functional interactor of SART3.

Please see also reply to Reviewer 3, point 4. We now include more data supporting the physical and functional interaction between *ALAL-1* and SART3.

6) Fig. 4 focuses on the transcriptional regulation of *ALAL-1* by NF- κ B. The authors need to show that this is independent of *IKBKB*-associated promoter and potential enhancer regions.

ALAL-1 and *IKBKB* are indeed antisense divergent RNAs sharing promoter and presumably other transcriptional regulatory genomic regions. As reported in several studies, antisense transcripts are widespread in all kingdoms of life and their expression can be regulated either co-ordinately or independently of their neighbouring genes (*Nature Reviews Genetics* volume 14, pages 880–893 (2013); *Nucleic Acids Res* 2017 Dec 1; 45(21): 12496–12508.; *Science* 02 Sep 2005:Vol. 309, Issue 5740, pp. 1564-1566). Global transcriptome analysis has shown that a large proportion of the genome can produce transcripts from both strands, arising from “neighbouring” transcriptional units and that frequent concordant regulation of sense/antisense pairs exists (*Science* 02 Sep 2005:Vol. 309, Issue 5740, pp. 1564-1566).

This situation of coordinated transcriptional regulation might well be the case for *ALAL-1* and *IKBKB*. To get more insight, we analysed the RNA levels of *ALAL-1* and its neighbour genes *IKBKB* and *PLAT* by qPCR in a TNF time course experiment and by RNA-seq of HCC95 treated or not with TNF for 4h (Fig.4C-D and Suppl.Fig.3B-D). Both approaches indicate a certain degree of co-regulation for the three genes since their transcription was similarly induced following 2-4h of TNF treatment. Other known NF- κ B target genes, such as *IL8* and *BCL3*, appeared instead as early-response genes, showing induction by TNF at 0.5-1h of treatment. Further experiments could clarify this aspect, however, as shown (Suppl.Fig.4 and response to Reviewer #1 point 1, Reviewer #3 point 3), *IKBKB* does not seem directly involved in the function of *ALAL-1* and we believe further studying the relationship between *ALAL-1* and its neighbour genes, even though interesting, goes behind the scope of this work.

Please see also response to Reviewer #1 point 1 and Reviewer #3, point 3.

7) The relationship between *ALAL-1*, SART3 and downstream pathways is largely correlative and needs further experimental support for functional interplay (Fig. 5).

We agree with the reviewer that in the previous version of the manuscript we just

showed correlative relationships. Now, thanks to the reviewers' suggestions, we provide new data that shade new light into the function of *ALAL-1* and its relationship with SART3. These new data indicate that, through its interaction with SART3, *ALAL-1* affects the subcellular localization of USP4, which in turn results in changes in the ubiquitination levels of protein components of the NF- κ B signalling pathway (Fig.6 and response to Reviewer #3). These changes in regulatory protein ubiquitination proteins are consistent with changes in signalling that result in indirect effects on gene expression, as reflected by the RNA-seq analyses.

These new results are detailed in the responses to Reviewer #3.

8) The initial evidence suggest that *ALAL-1* acts through cell autonomous mechanisms comes in conflict with the evidence for immune-modulatory phenotypes. Fig. 6 is weak and lacks depth, functional or mechanistic insights. The manuscript is perhaps better off without it.

The reviewer is correct that some of the assays employed can only address cell-autonomous effects. However, this is compatible with a dual *in vivo* phenotypic outcome. We propose that *ALAL-1* impacts tumour progression through both cell-autonomous and immune-modulatory mechanisms. On one hand, we show that the expression of *ALAL-1* promotes increased proliferation and viability of the tumour cell; on the other hand, it is able to modulate the tumour microenvironment. This is not unexpected since it is shown that *ALAL-1* regulates signalling pathways well known to activate intracellular mechanisms and the production of a number of cytokines that can affect the intrinsic proliferative state of the cell as well as the cell microenvironment. This is consistent with the data presented in Figure 7, based on the analysis of hundreds of patient samples, and suggesting a potential clinical relevance of our findings that we believe should be presented to the community. Nevertheless, we now are more careful in the wording of our conclusions to avoid overstatements.

Reviewer #3 (Comments to the Authors (Required)):

This is a study by Athie et al, describing identification and characterization of lncRNA, *ALAL-1* that is focally amplified in lung cancers. The authors analyzed somatic copy number alterations (SCNAs) using publicly available databases, detected the numerous SCNAs, and nailed down to previously uncharacterized *ALAL-1* lncRNA. *ALAL-1* is one of the targets of NF- κ B. *ALAL-1* regulates USP4 subcellular localization via tumor-rejection antigen SART3. Indeed, *ALAL-1* positively regulates proliferation of non-small lung cancer cell lines. In *ALAL-1* overexpressing lung cancer cell lines, the level of cytokines decreased, resulting in disabling a migration potency of several immune cells. With these observations, the authors suggest that *ALAL-1* is a proto-oncogenic lncRNA that mediates cancer immune evasion.

General comments:

Overall, the authors provide a potentially interesting concept that lung cancer cells overexpressing lncRNA ALAL-1 can evade host immune system. Their bioinformatic strategy to find new and interesting lncRNAs among SCNAs may be powerful. However, this is not novel, because a similar work identifying lncRNA genes in focal SCNAs among several types of cancers has been reported elsewhere (PMID:25203321).

We thank the reviewer for appreciating the value of our approach. He/she's correct that the analysis of SCNAs for the search of lncRNA drivers was applied in a previous study (Hu et al., 2014). However, that study was limited to only 2394 tumours, while here we include data derived from 7448 tumours, which increases significantly the power of our analysis.

A major concern is that the manuscript lacks convincing mechanistic insight into cellular function. The authors heavily depend on the bioinformatic results and previous knowledge, and does not fully and experimentally develop the main conclusion. Sometimes, the author's conclusion is over-stated. Besides, there are typos and mislabeling in the manuscript.

We thank the reviewer for his/her evaluation of our work and suggestions. We now present additional data that strengthen our conclusions.

Specific comments (major):

1. Interpretation of Figure 2C-E are confusing. I understand that ALAL-1 is highly expressed in the amplified group (Figure 2C) and in tumor (Figure 2D). However, majority of tumors (~ 66%) highly express ALAL-1 without the amplification of the ALAL-1 locus (Figure 2E). Moreover, ALAL-1 was also overexpressed in other types of tumors, such as LUSC, where ALAL-1 was not identified as frequently amplified (Supplementary Figure S2E). Therefore, importance of the ALAL-1 amplification in its overexpression and tumors is not clear.

First, the authors should not provide wrong impression that ALAL-1 is frequently amplified in cancer, which leads to its overexpression. A population of lung cancer with high expression and amplification of ALAL-1 is a minor (~7.2%) (Fig. 2E). They should describe the reason why they focus on this particular population.

Second, the authors should make it clear whether amplification of ALAL-1 gene is a cause of the high expression of ALAL-1.

The understanding of the reviewer is correct although finding the figure panels confusing.

The reviewer is correct that *ALAL-1* shows higher levels of expression in tumours when compared to normal samples in TCGA LUAD and other tumour cohorts, suggesting a pro-oncogenic role of the lncRNA dependent not only on its locus

amplification but also on the expression of the lncRNA. Tumours with *ALAL-1* amplification may well represent a specific subtype within LUAD cohort. Nevertheless, *ALAL-1* shows higher expression in LUAD samples, where our bioinformatics approach identified it as copy number-altered gene, in more than 81% of tumours that present the amplification, indicating that its higher expression level is linked to the amplification of the gene. Although apparently small, 7.2% of tumours presenting amplification of *ALAL-1* is a rather important proportion of tumours compared to other drivers. For instance, *EGFR*, one of the most recognized lung cancer drivers, is amplified in 9% of non-small-cell lung patients (Kato et al., 2019).

Most importantly, *ALAL-1* is overexpressed in both LUAD and LUASC cohorts independently of its amplified status, suggesting that the cancer cells activate different mechanisms that lead to *ALAL-1* increased expression. All these observations support the model that *ALAL-1* increased expression is advantageous for the tumour and thus linked to lung cancer.

2. In the abstract, the authors claim that *ALAL-1* is overexpressed through epigenetic mechanisms: "*ALAL-1* is also overexpressed in additional tumor types, such as lung squamous carcinoma through epigenetic mechanisms (line 7, page 2)". This is overstating. Bioinformatics analysis detected the two differentially methylated CpG sites (Figure 2 F-H). These show a correlation, but not enough evidence for "epigenetic mechanisms".

We have removed the sentence "*through epigenetic mechanisms*" from the abstract to lower the tone on this conclusion.

3. If the authors conclude that *ALAL-1* is a transcriptional target of NF- κ B, it should be tested by a promoter luciferase assay whether NF- κ B directly acts as a transcriptional factor towards *ALAL-1* expression in cultured cell lines. In addition, it is interesting to know whether NF- κ B expression also correlate with levels of *ALAL-1* expression in the patient samples.

As suggested, to experimentally test the biological activity of the NF- κ B regulatory elements, we cloned the *ALAL-1* most proximal genomic sequence (959 bp) containing the NF- κ B binding motif into a luciferase reporter vector and tested the reporter-gene induction with or without TNF- α treatment (Suppl.Fig.3E). As control, we also transfected cells with a commercially available NF- κ B reporter plasmid (pNF κ B-luc from Clontech). *ALAL-1* tested sequence was able to drive transcription of the reporter gene following TNF- α treatment. The induction observed in this experimental setting is of the same magnitude as the observed for the endogenous *ALAL-1* gene in a TNF- α and p65-dependent manner (Figure 4C and Supplementary Figures 3B). We believe that the body of evidence presented, which includes (i) induction of the endogenous *ALAL-1* upon TNF- α treatment (ii) decreased expression upon p65 knockdown (iii) presence of p65 motifs in *ALAL-1* (iv) binding to *ALAL-1* locus by endogenous p65 upon TNF- α treatment detected by ChIP-seq and (v) induction by TNF- α of a reporter gene by a putative p65 binding element of *ALAL-1*, strongly

support the conclusion that *ALAL-1* is transcriptionally controlled by NF- κ B.

As suggested by the Reviewer, we correlated the expression levels of *ALAL-1* and NF- κ B (*RELA*) mRNA in the TCGA LUAD and LUSC datasets observing no correlation in either lung tumour types (Reviewer Fig.3). The result is not unexpected since NF- κ B (*RELA*) and other members of the family of NF- κ B transcription factors are regulated predominantly at the post-translation level in the NF- κ B signalling pathway.

Reviewer Fig.3

Correlation of *ALAL-1* expression with NF- κ B (*RELA*) in TCGA LUAD and LUSC datasets.

4. The authors claim that the specific interaction between *ALAL-1* and *SART3* is crucial for the cellular localization of *USP4*, which potentially activate *USP4*'s function on the NF- κ B pathway. This argument is a key mechanistic insight of this manuscript, and should be addressed in more detail, as follows:

(4-1)

Although the authors detected *ALAL-1* predominantly in the cytoplasm, it is not still clear whether *ALAL-1* works in the nucleus or cytoplasm. Does *ALAL-1* and *SART3* make a complex mainly in the cytoplasm? In Fig.5B, the authors should perform the RIP assay using the cell lysates fractionated into the nucleus and cytoplasm, respectively. Also, they should address whether the level of this RNA-protein complex increase upon TNF- α .

(4-2)

Upon TNF- α -treatment, does the FISH signal of *ALAL-1* colocalize with that of HA-*SART3* in the cytoplasm?

To address these comments (4-1 and 4-2) we performed Co-Immuno-FISH experiments coupled to confocal microscopy that allow us to evaluate more precisely

and with high resolution the subcellular co-localization of SART3 and *ALAL-1*. SART3 localized both in the nucleus and cytoplasm, as observed by microscopy (Fig.6D and western blot (Fig.6E), while *ALAL-1* predominantly localized in the cytoplasm, with signal significantly increased with TNF treatment, as also previously observed (Fig.4E-F). Co-Immuno-FISH analysis showed clear localization of SART3 and *ALAL-1* in the cytoplasm (Fig.6D) while no co-localization was observed in the nucleus.

Similarly, Immuno-FISH experiments were also performed detecting USP4 and *ALAL-1*. USP4 localized both in the nucleus and cytoplasm, although most predominantly in the cytoplasm, as observed by microscopy (Fig.6F) and western blot (Fig.6G). Also in this case, *ALAL-1* predominantly localized in the cytoplasm and its signal increased when cells were treated with TNF (Fig.6F). Image analysis showed co-localization of USP4 and *ALAL-1* in the cytoplasm (Fig.6F). For both SART3 and USP4, the cytoplasmic co-localization with *ALAL-1* was clearer following TNF treatment (Fig.6D and F), suggesting that *ALAL-1* co-localize with SART3 or USP4 in the cytoplasm.

(4-3)

The authors found that the SART3-interaction region is located in the middle region of *ALAL-1* (Fig. 5D). Accordingly, the authors should create the mutant *ALAL-1* which does not associate with SART3. Then, (1) they should ask whether overexpression of this mutant version of *ALAL-1* fails to rescue the cellular localization of USP4, using the assay shown in Fig. 5F-G. (2) They should ask whether this mutant *ALAL-1* influence the expressions of cytokines. (3) They should also ask whether this mutant *ALAL-1* influences the proliferation of lung cancer cell lines (Supplementary Fig.4).

As also pointed by the Reviewer 2 (point 2), we addressed the ability of different fragments of *ALAL-1* - described in Fig.5D as able to interact with SART3 (fragment 6) or not (fragments 4 and 5) *in vitro* – in modulating the clonogenicity ability of cells. While the full-length version of *ALAL-1* was able to enhance cell clonogenicity, we did not observe any difference in colony formation when the truncated forms of *ALAL-1* were over expressed (Reviewer Fig.1), independently of their ability to bind SART3 *in vitro*. These data suggest that although some fragments of *ALAL-1* are able to interact with SART3, the interaction per se is not sufficient to recapitulate the function of *ALAL-1*. It is possible that other regions of *ALAL-1* contribute to the function of the lncRNA either by mediating additional molecular interactions and/or influencing the RNA structure. As addressed in reply to other points of the Reviewers, we have now added more evidences on the physical and functional interaction between *ALAL-1* and SART3-USP4 that strength our conclusions (Fig.6, Suppl.Fig.5F-H, Suppl.Fig.5M).

(4-4)

The authors also should make the mutant version of SART3 which cannot associate with *ALAL-1*, and ask whether overexpression of this mutant SART3 affect the cellular localization of USP4, using the assay shown in Fig.5G.

As previously reported by others (Park et al., 2016), SART3-USP4 interaction involves specific residues contained within the HAT10 and HAT11 repeat domains of SART3

(D497, M500, W511, Y514, R533) and not even the mutation of all five aminoacids was found able to abolish the interaction with USP4. Moreover, SART3 is known to play other roles involving interactions with its RNA recognition motifs (RRMs), e.g. it has a major role in the recycling phase of the spliceosome cycle by associating with U6 and U4/U6 snRNPs (Bell M. et al., 2002), as well as it has been reported to interact with RAD18 and Pol η through RRM1 in translesion DNA synthesis (Huang et al., 2018). Thus, it is expected that blocking *ALAL-1* binding by mutating SART3 will result in a general RNA binding impairment of SART3 that will eventually affect other functions and will not be informative to understand the specific role of *ALAL-1*.

(4-5)

Does depletion of *ALAL-1* disrupt the SART3-USP4 complex?

To address this possibility we first modelled the 3D structure of SART3, using the structural data described in Park et al. (2016, *Nucleic Acids Research*), and observed that SART3 binding to USP4 is compatible with RNA binding since the regions involved in these interactions are not in close proximity in the 3D space (Reviewer Figure 4). Additionally, we performed co-immunoprecipitation of SART3 and USP4 in cells depleted or not of *ALAL-1* by RNAi (Fig.6A), showing that the KD of *ALAL-1* does not affect the interaction between SART3 and USP4 and suggesting that the absence of *ALAL-1* does not impair the protein-protein interaction in bulk. Thus, based on these experimental data, subcellular co-localization and structural predictions, the *in vivo* interaction between SART3 and USP4 and *ALAL-1* is possible. Taking together all the data, we propose that the interaction between SART3 and *ALAL-1* regulates the subcellular localization of USP4 by promoting its translocation to the nucleus.

Reviewer Fig.4

Whole SART3 3D modelling according to the structural data published by Park et al. (2016, *Nucleic Acids Research*), showing the domains interacting with PRPF3 (green), USP4 (blue), Lsm (yellow) and RNA (red).

(4-6)

Upon depletion of ALAL-1, the SART3-USP4 complex does not localize properly (Fig.5F). Does this mean that this complex would not exert its deubiquitination ability toward the substrates upon depletion of ALAL-1? If so, the authors should show one or more substrates (factors involved in the NF- κ B pathways) whose poly-(or mono-) ubiquitination decreased upon ALAL-1-depletion.

We thank the reviewer for the experimental suggestion and now include data obtained analysing the deubiquitination ability of USP4 in the presence or absence of ALAL-1. For this, we selected two previously reported substrates of USP4, PRP3 (Song EJ et al., 2010) and TAK1 (Fan YH et al., 2011), and analysed their ubiquitination in the presence or absence of ALAL-1. Interestingly, while depletion of ALAL-1 by RNAi increased the levels of the ubiquitinated forms of PRP3 (Fig.6C), the amount of ubiquitinated TAK1 was reduced (Fig.6B). With the notion that PRP3 and TAK1 are USP4 substrates predominantly nuclear and cytoplasmatic respectively, the results obtained seem to support our previous data and model that sees ALAL-1 depletion causing a higher cytoplasmatic localization of USP4, in turn expected to cause a higher deubiquitination of its cytoplasmatic substrate TAK1 and a lower deubiquitination of its nuclear substrate PRP3.

In summary, our data indicate that ALAL-1 is required for the increased nuclear localization of USP4 and subsequent increased ubiquitination of its cytoplasmic substrate TAK1. Based on our own data and the studies published by other groups, we propose that ALAL-1 induces conformational changes in SART3/USP4 that promote USP4 translocation, for which importin- α has been shown to be required (Park et al., 2016). Although structural studies are needed to shed light into the mechanistic details of this phenomenon, it is clear that the transcriptional activation of ALAL-1 in response to TNF α , and its subsequent association with SART3, allows the coordination between these factors to induce an increased nuclear localization of USP4.

5. In Fig.5C, it lacks control experiment using RNase, which can show back ground levels of this experiment.

We respectfully disagree with the reviewer's suggestion, since we do not believe it will be informative. RIP experiment detects RNA, so if we treat the samples with RNase the signal (both specific and background) will be strongly reduced, but this won't give any additional information of the background or specificity. In fact, that type of control is never used in RIP experiments. On the other hand, an unrelated antibody (such as IgG) can give a good idea of the background levels.

6. The authors claim that ALAL-1 is pro-oncogene. It will be nice if the authors can show its oncogenic effects of ALAL-1 in mice, by xenograft of ALAL-1 overexpressing cell lines.

As suggested by the reviewer we analyzed the pro-oncogenic ability of *ALAL-1 in vivo* with xenograft models. For this, we used A549 cells stably overexpressing *ALAL-1* and

by comparing their ability to form tumors in mice with WT or control cells (empty vector), we observed a significant increase in tumor volumes in mice injected with *ALAL-1* OE cells (Suppl.Fig.2T), thus supporting the results obtained by depletion of *ALAL-1* (Fig.3J and M) and the oncogenic role of the lncRNA.

Specific comments (minor):

1. The authors should carefully cite appropriate references in the text. On page 10, in the last paragraph, ref. #32, 33, 41, 42 and 43 are cited in wrong place.

We have corrected it. The mistake was due to the EndNote Bibliography formatting.

2. On page 10, lines 3 and 5, wrong figure numbers are referred (Supplementary Fig. 5E, instead of 5C).

We have corrected it in the text.

3. On page 10, lines 28 and 29, wrong figure numbers are referred (Figure 6A-B).

We have corrected it in the text.

4. I could not understand why the agarose gel in Supplementary Fig. 3B (right, "OUT") does not show the parental slow-migrating band. If the deletion by CRISPR was heterozygous in clones 23 and 24, as the authors described in the fourth paragraph on page 7, the clones must contain both the deleted and intact alleles.

Although very weak, a band at the size of the WT allele was visible, shown below in the contrast adjusted images of two different gels. Our reasoning is that, due to the smaller size of the edited allele, the PCR reaction over this fragment is much more efficient than over the wild type, showing a clear bias in the final product.

Reviewer Fig.5. Contrast adjusted images of the PCR products obtained for the CRISPR clones screening.

5. There are typos in the Figure legends, as follows.

- ...de lncRNA PR11..... (Figure 1 G, on page 19, line 16)
- CAN 623 should be read as CNA623 (Figure 2A, on page 19, line 22).
- "...in read." should be read as "...in red." (Figure 2A, on page 19, line 23) .

- "...exon D..." should be read as "...exon 3..."(Figure 3E, on page 20, line 14)
- The sentence "...HUVEC cells treated or and treated with... should be corrected. (Figure 4A, on page 21, line 1)
- The sentence "ALAL-1 interacts with regulates key cellular key cellular...." should be corrected (Figure 5, on page 21, line 16)
- "E. Number of cells of..." is incorrectly labeled as "C. Number of cells of..." (Figure 6, on page 22, line 11).
- "...500 bp flanking exon D..." should be read as "...500 bp flanking exon 3..." (in the legend for Supplementary Figure 3A).

We have corrected the typos in the text.

6. There are possible typos in the Method section, as follows.

- In the text and the Figure 6 legend, the authors describe that they used A549 cells for the cell migration and cytokine production assays (lines 31 and 33 on page 11, and lines 9 and 11 on page 22). Regardless, they describe that they used HCC95 cells, in the Method section. They have to correct the discrepancy, for the sake of potential readers.
- In the Immunofluorescence section, the antibodies detecting HA tag (or the endogenous SART3?) used for Figure 5F should be described.

We have corrected the typos in the text and included information on the ab.

REFERENCES:

- Hu, X., Y. Feng, D. Zhang, S.D. Zhao, Z. Hu, J. Greshock, Y. Zhang, L. Yang, X. Zhong, L.P. Wang, S. Jean, C. Li, Q. Huang, D. Katsaros, K.T. Montone, J.L. Tanyi, Y. Lu, J. Boyd, K.L. Nathanson, H. Li, G.B. Mills, and L. Zhang. 2014. A functional genomic approach identifies FAL1 as an oncogenic long noncoding RNA that associates with BMI1 and represses p21 expression in cancer. *Cancer Cell*. 26:344-357.
- Kato, S., R. Okamura, M. Mareboina, S. Lee, A. Goodman, S.P. Patel, P.T. Fanta, R.B. Schwab, P. Vu, V.M. Raymond, R.B. Lanman, J.K. Sicklick, S.M. Lippman, and R. Kurzrock. 2019. Revisiting Epidermal Growth Factor Receptor (EGFR) Amplification as a Target for Anti-EGFR Therapy: Analysis of Cell-Free Circulating Tumor DNA in Patients With Advanced Malignancies. *JCO Precis Oncol*. 3.
- Park, J.K., T. Das, E.J. Song, and E.E. Kim. 2016. Structural basis for recruiting and shuttling of the spliceosomal deubiquitinase USP4 by SART3. *Nucleic Acids Res*. 44:5424-5437.
- Song, E.J., S.L. Werner, J. Neubauer, F. Stegmeier, J. Aspden, D. Rio, J.W. Harper, S.J. Elledge, M.W. Kirschner, and M. Rape. 2010. The Prp19 complex and the Usp4Sart3 deubiquitinating enzyme control reversible ubiquitination at the spliceosome. *Genes Dev*. 24:1434-1447.
- Fan, Y.H., Y. Yu, R.F. Mao, X.J. Tan, G.F. Xu, H. Zhang, X.B. Lu, S.B. Fu, and J. Yang. 2011. USP4 targets TAK1 to downregulate TNFalpha-induced NF-kappaB activation. *Cell Death Differ*. 18:1547-1560
- Bell, M., S. Schreiner, A. Damianov, R. Reddy, and A. Bindereif. 2002. p110, a novel human U6 snRNP protein and U4/U6 snRNP recycling factor. *EMBO J*. 21:2724-2735.
- Huang, M., B. Zhou, J. Gong, L. Xing, X. Ma, F. Wang, W. Wu, H. Shen, C. Sun, X. Zhu, Y. Yang, Y. Sun, Y. Liu, T.S. Tang, and C. Guo. 2018. RNA-splicing factor SART3 regulates translesion DNA synthesis. *Nucleic Acids Res*. 46:4560-4574.

February 24, 2020

Re: JCB manuscript #201908078R

Dr. Maite Huarte
CIMA
Pio XII, 55
Pamplona 31008
Spain

Dear Dr. Huarte,

Thank you for submitting your revised manuscript entitled "Copy number alterations from a lncRNA perspective reveal a regulator of lung cancer immune evasion". We apologize for the extensive delay in providing you with a decision.

In any case, the manuscript has been seen by the original reviewers whose full comments are appended below. While the reviewers continue to be overall positive about the work in terms of its suitability for JCB, some important issues remain.

You will see that although reviewers #1 and #3 now recommend acceptance, reviewer #2 continues to raise a number of largely substantive issues that we feel need to be addressed before the paper will be ready for publication. Below, we list each of this reviewer's specific comments and provide our thoughts on how best to address them:

1) "The authors did not sufficiently address the role of IKBKB in human cancer and in the processes that they are studying. The authors continue to stress the misleading notion that their manuscript is about genetic alterations that "lie in regions of the genome devoid of protein-coding genes" (Second sentence of Introduction) and that they have "identified lncRNAs frequently lost or amplified independently of protein-coding genes" (Abstract). Clearly, that is not the case of ALAL-1, which is the main focus of this study. The persistent evasion of the fact that ALAL-1 is both co-regulated and co-amplified with its neighboring gene, IKBKB (as stated by the authors in Rebuttal) is very concerning."

- We feel that this point can be addressed by further clarification and attempting to better frame your findings without overstating them.

2) "The authors have proposed a model where IKBKB and ALAL-1 function in independent, possibly parallel, pathways. However, they have not provided support for this model, which is essential to their study (other than there is one instance of a peak that only contains ALAL-1). If it turns out that IKBKB is the oncogene in the locus (and drives locus amplification, increased expression from the locus, and locus methylation), then the impact of their findings would be significantly diminished, regardless of what the in vitro data show."

- While we agree that this is an interesting point, we feel that you have already addressed part of this concern by showing that perturbation of ALAL-1 does not affect the expression of IKBKB. However, this does not completely address the problem of epistasis, raised by the reviewer in the

original round of critique. To do this, you should also show that by perturbing IKBKB expression, the levels of ALAL-1 are not altered. This would provide further support for your proposed parallel and independent pathway hypothesis (by excluding the possibility that ALAL-1 acts downstream of IKBKB).

3) "The authors did not perform the full set of rescue experiment, including in wild-type clones (the reviewer assumes they used wt clones and not the parental population in these experiments) and deletion clones (23+24) as well as in RNAi-treated samples. Whether or not lncRNAs are functional molecules remains an open question, despite a large number of published works that can be cited, in part because many of these publications do not report reproducible findings. This has led to increased stringency in the requirements to show a role for "RNA product" (as stated by authors in Abstract). This manuscript does not show a convincing role for the RNA product. The rescue experiments are not a complete set, show in some cases small and in other excessively large effects, and the data are difficult to navigate and interpret. In addition, RNA FISH images are poor quality (Figure 4F and G, images are scored at 150 foci per cell). This was a key opportunity for the authors to provide support for the validity of ALAL-1 as a functional gene."

- While we see this reviewer's point, we feel that the CRISPR-based deletion experiments already provided adequately address much of this issue. However, this issue of the functional meaning of the interaction between ALAL1 and SART3 is important to address. Here, we agree that the provided experiments fall short of the target in two instances: i) lack of rescue by mutant ALAL-1 is not shown. This is particularly relevant for mutants lacking the SART3 binding domain (as expressly requested by rev. 2 in the first round), ii) in point 5 of the first review, rev. 2 asks: "The authors need to provide further experimental evidence that the growth and colony formation phenotypes are mediated through SART3"; the experiments performed in response to this are not completely adequate. The authors should directly tackle the issue by using ALAL-1 mutants without the SART3-binding surface". We understand that this might be problematic because the SART3 binding surface is only roughly mapped, but this is an issue that requires further efforts. Some of the results might be already contained in the Fig. 1 for reviewer, attached to the rebuttal...yet we could not understand exactly what that figure depicts, since "fragments" not mentioned in the paper (fragment 7) are used (or mislabeled).

Finally, better quality pictures should be provided, as per the reviewer's request.

4) "The statement of the importance of ALAL-1 in TNF, p53, NF- κ B and TGF- β pathways (Abstract) is a direct overstatement that has not been shown in any way by the authors (besides GO analysis).

The explanation provided (in the Rebuttal) for the dual tumor intrinsic and extrinsic roles is not sufficient in the absence of direct experimental evidence."

- These final two issues can be addressed with changes to the text, toning down your conclusions somewhat.

Our general policy is that papers are considered through only one revision cycle; however, given that the suggested changes are relatively minor we are open to one additional short round of revision.

Please submit the final revision within one to two months, along with a cover letter that includes a point by point response to the remaining reviewer comments.

Thank you for this interesting contribution to Journal of Cell Biology. You can contact me or the

scientific editor listed below at the journal office with any questions, cellbio@rockefeller.edu or call (212) 327-8588.

Sincerely,

Pier Paolo Di Fiore, MD, PhD
Senior Editor
The Journal of Cell Biology

Tim Spencer, PhD
Executive Editor
Journal of Cell Biology

Reviewer #1 (Comments to the Authors (Required)):

The authors have adequately responded to my previously raised concerns and provide additional evidence supporting their conclusions as well as clarifications.

Reviewer #2 (Comments to the Authors (Required)):

While the authors have added a number of experiments, the key concerns with the manuscript were not addressed. The manuscript remains rather expansive and does not provide a solid and reliable contribution to the field about the role and significance of ALAL-1. I would recommend rejection of the manuscript.

Below, key points are summarized, in particular as they relate to statements made by the authors in the Abstract.

- The authors did not sufficiently address the role of IKBKB in human cancer and in the processes that they are studying. The authors continue to stress the misleading notion that their manuscript is about genetic alterations that "lie in regions of the genome devoid of protein-coding genes" (Second sentence of Introduction) and that they have "identified lncRNAs frequently lost or amplified independently of protein-coding genes" (Abstract). Clearly, that is not the case of ALAL-1, which is the main focus of this study. The persistent evasion of the fact that ALAL-1 is both co-regulated and co-amplified with its neighboring gene, IKBKB (as stated by the authors in Rebuttal) is very concerning.
- The authors have proposed a model where IKBKB and ALAL-1 function in independent, possibly parallel, pathways. However, they have not provided support for this model, which is essential to their study (other than there is one instance of a peak that only contains ALAL-1). If it turns out that IKBKB is the oncogene in the locus (and drives locus amplification, increased expression from the locus, and locus methylation), then the impact of their findings would be significantly diminished, regardless of what the in vitro data show.
- The authors did not perform the full set of rescue experiment, including in wild-type clones (the reviewer assumes they used wt clones and not the parental population in these experiments) and deletion clones (23+24) as well as in RNAi-treated samples. Whether or not lncRNAs are functional molecules remains an open question, despite a large number of published works that can be cited, in part because many of these publications do not report reproducible findings. This has led to increased stringency in the requirements to show a role for "RNA product" (as stated by authors in Abstract). This manuscript does not show a convincing role for the RNA product. The rescue

experiments are not a complete set, show in some cases small and in other excessively large effects, and the data are difficult to navigate and interpret. In addition, RNA FISH images are poor quality (Figure 4F and G, images are scored at 150 foci per cell). This was a key opportunity for the authors to provide support for the validity of ALAL-1 as a functional gene.

- The statement of the importance of ALAL-1 in TNF, p53, NF-kB and TGF- β pathways (Abstract) is a direct overstatement that has not been shown in any way by the authors (besides GO analysis).
- The explanation provided (in the Rebuttal) for the dual tumor intrinsic and extrinsic roles is not sufficient in the absence of direct experimental evidence.

Reviewer #3 (Comments to the Authors (Required)):

The authors have performed additional experiments to address most of the critiques raised by the reviewers. They had clarified the concerns, by providing the explanations in the rebuttal letter, and revising the text appropriately.

2nd Revision - Authors' Response to Reviewers: May 8, 2020

Dear Dr. Huarte,

Thank you for submitting your revised manuscript entitled "Copy number alterations from a lncRNA perspective reveal a regulator of lung cancer immune evasion". We apologize for the extensive delay in providing you with a decision.

In any case, the manuscript has been seen by the original reviewers whose full comments are appended below. While the reviewers continue to be overall positive about the work in terms of its suitability for JCB, some important issues remain.

You will see that although reviewers #1 and #3 now recommend acceptance, reviewer #2 continues to raise a number of largely substantive issues that we feel need to be addressed before the paper will be ready for publication. Below, we list each of this reviewer's specific comments and provide our thoughts on how best to address them:

1) "The authors did not sufficiently address the role of IKBKB in human cancer and in the processes that they are studying. The authors continue to stress the misleading notion that their manuscript is about genetic alterations that "lie in regions of the genome devoid of protein-coding genes" (Second sentence of Introduction) and that they have "identified lncRNAs frequently lost or amplified independently of protein-coding genes" (Abstract). Clearly, that is not the case of ALAL-1, which is the main focus of this study. The persistent evasion of the fact that ALAL-1 is both co-regulated and co-amplified with its neighboring gene, IKBKB (as stated by the authors in Rebuttal) is very concerning."

- We feel that this point can be addressed by further clarification and attempting to better frame your findings without overstating them.

ALAL-1 was identified because the consensus focal peak determined by GISTIC algorithm only affected the lncRNA (Figure 2). Nevertheless, given the size of amplified genomic regions, we acknowledge (and state in the text) that in individual tumours ALAL-1 and IKBKB (as well as other genes from the same chromosomal region) may be co-amplified. Nevertheless, we believe that we provide sufficient evidence to show that ALAL-1 has inherent pro-oncogenic function.

Cellular networks are highly complex, and we acknowledge that functional interactions between ALAL-1 and other co-expressed genes probably exist (including IKBKB and many others). However their characterization go beyond the scope of this manuscript.

We are very careful with the phrasing throughout the manuscript, so we do not overstate our findings. We have now removed from the abstract "independently from protein-coding genes".

2) "The authors have proposed a model where IKBKB and ALAL-1 function in independent, possibly parallel, pathways. However, they have not provided support for this model, which is essential to their study (other than there is one instance of a peak that only contains ALAL-1). If it turns out that IKBKB is the oncogene in the locus (and drives locus amplification, increased expression from the locus, and locus methylation), then the impact of their findings would be significantly diminished, regardless of what the in vitro data show."

- While we agree that this is an interesting point, we feel that you have already addressed part of this concern by showing that perturbation of ALAL-1 does not affect the expression of IKBKB. However, this does not completely address the problem of epistasis, raised by the reviewer in the original round of critique. To do this, you should also show that by perturbing IKBKB expression, the levels of ALAL-1 are not altered. This would provide further support for your proposed parallel and independent pathway hypothesis (by excluding the possibility that ALAL-1 acts downstream of IKBKB).

Following the editors' suggestion, we performed experiments where we deplete cells of ALAL-1 or IKBKB by RNAi and measured the levels of either RNAs. As expected, depletion of *IKBKB* by RNAi reduced the level of ALAL-1 (new Supplementary Fig. 4G), in line with the notion that ALAL-1 is a transcriptional target of NF- κ B, and IKBKB is the kinase that regulates the inhibitor/NF- κ B complex. The results are therefore in agreement

with our other data showing ALAL-1 as a transcriptional target of NF- κ B. Indeed, depletion of IKBKB also reduced the levels of IL6, well-known target of the NF- κ B pathway, but with inherent molecular functions independent of IKBKB. However, side by side depletion of ALAL-1 did not reduce the levels of IL6, suggesting that although both genes are co-regulated by NF- κ B, the effects observed upon ALAL-1 depletion are not driven by *IKBKB*, and that the two function independently.

3) "The authors did not perform the full set of rescue experiment, including in wild-type clones (the reviewer assumes they used wt clones and not the parental population in these experiments) and deletion clones (23+24) as well as in RNAi-treated samples. Whether or not lncRNAs are functional molecules remains an open question, despite a large number of published works that can be cited, in part because many of these publications do not report reproducible findings. This has led to increased stringency in the requirements to show a role for "RNA product" (as stated by authors in Abstract). This manuscript does not show a convincing role for the RNA product. The rescue experiments are not a complete set, show in some cases small and in other excessively large effects, and the data are difficult to navigate and interpret. In addition, RNA FISH images are poor quality (Figure 4F and G, images are scored at 150 foci per cell). This was a key opportunity for the authors to provide support for the validity of ALAL-1 as a functional gene."

- While we see this reviewer's point, we feel that the CRISPR-based deletion experiments already provided adequately address much of this issue. However, this issue of the functional meaning of the interaction between ALAL1 and SART3 is important to address. Here, we agree that the provided experiments fall short of the target in two instances: i) lack of rescue by mutant ALAL-1 is not shown. This is particularly relevant for mutants lacking the SART3 binding domain (as expressly requested by rev. 2 in the first round), ii) in point 5 of the first review, rev. 2 asks: "The authors need to provide further experimental evidence that the growth and colony formation phenotypes are mediated through SART3"; the experiments performed in response to this are not completely adequate. The authors should directly tackle the issue by using ALAL-1 mutants without the SART3-binding surface". We understand that this might be problematic because the SART3 binding surface is only roughly mapped, but this is an issue that requires further efforts. Some of the results might be already contained in the Fig. 1 for reviewer, attached to the rebuttal...yet we could not understand exactly what that figure depicts, since "fragments" not mentioned in the paper (fragment 7) are used (or mislabeled).

Finally, better quality pictures should be provided, as per the reviewer's request.

We have performed new clonogenicity experiments where we transfect HCC95 cells with an empty vector as control or a vector containing different ALAL-1 constructs, i.e. a full-length ALAL-1, a non-binding and a SART3-binding fragment, fragment #5 and #6 respectively (schematic in Figure 5D). Moreover, cells were depleted or not of SART3 by RNAi. The new results are shown in **Supplementary Fig. 5D**.

As previously observed with several cell lines (**Supplementary Fig. 2K-S**), overexpression of ALAL-1 full-length increased the clonogenic capacity of HCC95 cells (**Supplementary Fig. 5D**) and interestingly, this effect is impaired by depletion of SART3, strengthening our conclusion that colony formation phenotypes specific of ALAL-1 overexpression are mediated through SART3. As for the non-binding and SART3-binding fragments of ALAL-1 we did not observe any effect in terms of clonogenic capacity independently of their ability to bind SART3 in vitro (**Supplementary Fig. 5D**). We hypothesise that in vivo ALAL-1 truncations are not sufficient to induce a phenotype and that additional regions of the lncRNA may be necessary for ALAL-1 cellular activity.

The RNA FISH images shown in **Figure 4F** have been replaced for better quality images. Experiments were repeated and images acquired with confocal microscopy.

4) "The statement of the importance of ALAL-1 in TNF, p53, NF- κ B and TGF- β pathways (Abstract) is a direct overstatement that has not been shown in any way by the authors (besides GO analysis). The explanation provided (in the Rebuttal) for the dual tumor intrinsic and extrinsic roles is not sufficient in the absence of direct experimental evidence."

- These final two issues can be addressed with changes to the text, toning down your conclusions

somewhat.

We have rephrased the sentence in the abstract to: *"regulating the subcellular localization of the protein deubiquitinase USP4 and in turn, its function in the cell"*.

Our general policy is that papers are considered through only one revision cycle; however, given that the suggested changes are relatively minor we are open to one additional short round of revision.

Please submit the final revision within one to two months, along with a cover letter that includes a point by point response to the remaining reviewer comments.

Thank you for this interesting contribution to Journal of Cell Biology. You can contact me or the scientific editor listed below at the journal office with any questions, cellbio@rockefeller.edu or call (212) 327-8588.

Sincerely,

Pier Paolo Di Fiore, MD, PhD
Senior Editor
The Journal of Cell Biology

Tim Spencer, PhD
Executive Editor
Journal of Cell Biology

Reviewer #1 (Comments to the Authors (Required)):

The authors have adequately responded to my previously raised concerns and provide additional evidence supporting their conclusions as well as clarifications.

Reviewer #2 (Comments to the Authors (Required)):

While the authors have added a number of experiments, the key concerns with the manuscript were not addressed. The manuscript remains rather expansive and does not provide a solid and reliable contribution to the field about the role and significance of ALAL-1. I would recommend rejection of the manuscript. Below, key points are summarized, in particular as they relate to statements made by the authors in the Abstract.

- The authors did not sufficiently address the role of IKBKB in human cancer and in the processes that they are studying. The authors continue to stress the misleading notion that their manuscript is about genetic alterations that "lie in regions of the genome devoid of protein-coding genes" (Second sentence of Introduction) and that they have "identified lncRNAs frequently lost or amplified independently of protein-coding genes" (Abstract). Clearly, that is not the case of ALAL-1, which is the main focus of this study. The persistent evasion of the fact that ALAL-1 is both co-regulated and co-amplified with its neighboring gene, IKBKB (as stated by the authors in Rebuttal) is very concerning.
- The authors have proposed a model where IKBKB and ALAL-1 function in independent, possibly parallel, pathways. However, they have not provided support for this model, which is essential to their study (other than there is one instance of a peak that only contains ALAL-1). If it turns out that IKBKB is the oncogene in the locus (and drives locus amplification, increased expression from the locus, and locus methylation), then the impact of their findings would be significantly diminished, regardless of what the in vitro data show.
- The authors did not perform the full set of rescue experiment, including in wild-type clones (the reviewer assumes they used wt clones and not the parental population in these experiments) and deletion clones (23+24) as well as in RNAi-treated samples. Whether or not lncRNAs are functional molecules remains an open question, despite a large number of published works that can be cited, in part because many of these publications do not report reproducible findings. This has led to increased stringency in the requirements to

show a role for "RNA product" (as stated by authors in Abstract). This manuscript does not show a convincing role for the RNA product. The rescue experiments are not a complete set, show in some cases small and in other excessively large effects, and the data are difficult to navigate and interpret. In addition, RNA FISH images are poor quality (Figure 4F and G, images are scored at 150 foci per cell). This was a key opportunity for the authors to provide support for the validity of ALAL-1 as a functional gene.

- The statement of the importance of ALAL-1 in TNF, p53, NF- κ B and TGF- β pathways (Abstract) is a direct overstatement that has not been shown in any way by the authors (besides GO analysis).
- The explanation provided (in the Rebuttal) for the dual tumor intrinsic and extrinsic roles is not sufficient in the absence of direct experimental evidence.

Reviewer #3 (Comments to the Authors (Required)):

The authors have performed additional experiments to address most of the critiques raised by the reviewers. They had clarified the concerns, by providing the explanations in the rebuttal letter, and revising the text appropriately.

May 20, 2020

RE: JCB Manuscript #201908078RR

Dr. Maite Huarte
CIMA
Pio XII, 55
Pamplona 31008
Spain

Dear Dr. Huarte:

Thank you for submitting your revised manuscript entitled "The analysis of copy number alterations from a lncRNA perspective reveal a regulator of lung cancer immune evasion". We have now assessed your revised paper and we would be happy to publish your paper in JCB pending some final revisions to both the content and in order to meet our formatting guidelines (see details below).

There are two main issues that will need to be addressed in the final revision.

First, regarding your statement:

"Of note, ALAL-1 is localized upstream of IKBKB, known NF- κ B target gene (Supplementary Fig. 4A). While IKBKB is co-amplified with ALAL-1 in some tumors, due to their genomic proximity (Supplementary Fig. 4B), we did not observe any effect on IKBKB expression upon ALAL-1 truncation by CRISPR nor by siRNA knockdown (Supplementary Fig. 4C-F). Depletion of IKBKB by RNAi reduced instead the level of ALAL-1 (Supplementary Fig. 4G), in line with the notion that ALAL-1 is a transcriptional target of NF- κ B. These data suggest that, although both genes are coregulated by NF- κ B, the effects observed upon ALAL-1 depletion are not driven by IKBKB, indicating that ALAL-1 functions independently of IKBKB".

Based on the shown experiments, we do not feel that you can fairly conclude that "ALAL-1 functions independently of IKBKB". The data show a complex network of reciprocal interactions between the components of the pathway. It is likely that you may be referring to the fact that if one looks at the circuitry from the cancer perspective, the selective advantage conferred by ALAL-1 is independent of IKBKB. On this we agree. However, as presently formulated, the sentence has "physiological" implications about the circuitry that are unwarranted. Thus, you will need to modify your conclusions to make it clear that you are not drawing conclusions about the physiological working of the circuitry but only on its impact on cancer phenotypes.

Next, in the discussion you state:

"Although this strategy may result in the loss of lncRNAs that act in coordination with other gene loci, it led us to identify ALAL-1, a lncRNA with bona-fide oncogenic features."

This is a bit awkward. A gene is either a bona-fide oncogene or it is not. We do not think that this manuscript is sufficient to establish ALAL-1 as a bona-fide oncogene. However it is sufficient to say that it displays oncogenic features. Thus, the word "bona-fide" should be removed.

**Be sure to include a rebuttal with your final revision which illustrates how you addressed these

points.**

A. MANUSCRIPT ORGANIZATION AND FORMATTING:

Full guidelines are available on our Instructions for Authors page, <http://jcb.rupress.org/submission-guidelines#revised>. **Submission of a paper that does not conform to JCB guidelines will delay the acceptance of your manuscript.**

- 1) Text limits: Character count for Articles and Tools is < 40,000, not including spaces. Count includes title page, abstract, introduction, results, discussion, and acknowledgments. Count does not include materials and methods, figure legends, references, tables, or supplemental legends. You are currently below this limit but please bear it in mind when revising.
- 2) Figure formatting: Scale bars must be present on all microscopy images, including inset magnifications. Therefore, you will need to add scale bars to the images in figures 4F,5F, 6D, and 6F. Molecular weight or nucleic acid size markers must be included on all gel electrophoresis, including cropped images. Please add these markers to the gels/blots in figures 4B, 5B, 5D, 6A, 6B, 6C, 6E, 6G, S2B, S5J, and S5K.
- 3) Statistical analysis: Error bars on graphic representations of numerical data must be clearly described in the figure legend. The number of independent data points (n) represented in a graph must be indicated in the legend. Statistical methods should be explained in full in the materials and methods. For figures presenting pooled data the statistical measure should be defined in the figure legends. Please also be sure to indicate the statistical tests used in each of your experiments (both in the figure legend itself and in a separate methods section) as well as the parameters of the test (for example, if you ran a t-test, please indicate if it was one- or two-sided, etc.). Also, if you used parametric tests, please indicate if the data distribution was tested for normality (and if so, how). If not, you must state something to the effect that "Data distribution was assumed to be normal but this was not formally tested."
- 4) Title: The title should be concise but accessible to a general readership. While your current title will be appreciated by the specialists, we do not feel that it will be accessible to a broader cell biology audience. Therefore, please change your title to the following: "Analysis of copy number alterations reveal the lncRNA ALAL-1 as a regulator of lung cancer immune evasion"
- 5) Materials and methods: Should be comprehensive and not simply reference a previous publication for details on how an experiment was performed. Please provide full descriptions (at least in brief) in the text for readers who may not have access to referenced manuscripts. The text should not refer to methods "...as previously described."
Please also note that the materials and methods should be included in the main text and not in the supplement.
- 6) Please be sure to provide the sequences for all of your primers/oligos and RNAi constructs in the materials and methods. You must also indicate in the methods the source, species, and catalog numbers (where appropriate) for all of your antibodies. Please note that we cannot accommodate tables in the methods section. Therefore, we recommend that you either list your primers/sgRNAs/siRNAs in sentence form or make them into separate supplementary tables and

refer to them in the methods.

7) Microscope image acquisition: The following information must be provided about the acquisition and processing of images:

- a. Make and model of microscope
- b. Type, magnification, and numerical aperture of the objective lenses
- c. Temperature
- d. imaging medium
- e. Fluorochromes
- f. Camera make and model
- g. Acquisition software
- h. Any software used for image processing subsequent to data acquisition. Please include details and types of operations involved (e.g., type of deconvolution, 3D reconstitutions, surface or volume rendering, gamma adjustments, etc.).

8) References: There is no limit to the number of references cited in a manuscript. References should be cited parenthetically in the text by author and year of publication. Abbreviate the names of journals according to PubMed.

****Note, however, that we do not allow a separate reference section in the supplementary materials. Therefore, you must remove the reference list at the end of your methods section and incorporate any non-duplicated references into the main reference list.****

9) Supplemental materials: There are strict limits on the allowable amount of supplemental data. Articles/Tools may have up to 5 supplemental figures. At the moment, you are below this limit but please bear it in mind when revising.

Please also note that tables, like figures, should be provided as individual, editable files. A summary of all supplemental material should appear at the end of the Materials and methods section.

10) Conflict of interest statement: JCB requires inclusion of a statement in the acknowledgements regarding competing financial interests. If no competing financial interests exist, please include the following statement: "The authors declare no competing financial interests." If competing interests are declared, please follow your statement of these competing interests with the following statement: "The authors declare no further competing financial interests."

11) A separate author contribution section is required following the Acknowledgments in all research manuscripts. All authors should be mentioned and designated by their first and middle initials and full surnames. We encourage use of the CRediT nomenclature (<https://casrai.org/credit/>).

12) ORCID IDs: ORCID IDs are unique identifiers allowing researchers to create a record of their various scholarly contributions in a single place. At resubmission of your final files, please consider providing an ORCID ID for as many contributing authors as possible.

B. FINAL FILES:

-- High-resolution figure and video files: See our detailed guidelines for preparing your production-ready images, <http://jcb.rupress.org/fig-vid-guidelines>.

Thank you for this interesting contribution, we look forward to publishing your paper in Journal of Cell Biology.

Sincerely,

Pier Paolo Di Fiore, MD, PhD
Senior Editor
The Journal of Cell Biology

Tim Spencer, PhD
Executive Editor
Journal of Cell Biology

Dear Dr. Huarte:

Thank you for submitting your revised manuscript entitled "The analysis of copy number alterations from a lncRNA perspective reveal a regulator of lung cancer immune evasion". We have now assessed your revised paper and we would be happy to publish your paper in JCB pending some final revisions to both the content and in order to meet our formatting guidelines (see details below).

There are two main issues that will need to be addressed in the final revision.

First, regarding your statement:

"Of note, ALAL-1 is localized upstream of IKBKB, known NF- κ B target gene (Supplementary Fig. 4A). While IKBKB is co-amplified with ALAL-1 in some tumors, due to their genomic proximity (Supplementary Fig. 4B), we did not observe any effect on IKBKB expression upon ALAL-1 truncation by CRISPR nor by siRNA knockdown (Supplementary Fig. 4C-F). Depletion of IKBKB by RNAi reduced instead the level of ALAL-1 (Supplementary Fig. 4G), in line with the notion that ALAL-1 is a transcriptional target of NF- κ B. These data suggest that, although both genes are coregulated by NF- κ B, the effects observed upon ALAL-1 depletion are not driven by IKBKB, indicating that ALAL-1 functions independently of IKBKB".

Based on the shown experiments, we do not feel that you can fairly conclude that "ALAL-1 functions independently of IKBKB". The data show a complex network of reciprocal interactions between the components of the pathway. It is likely that you may be referring to the fact that if one looks at the circuitry from the cancer perspective, the selective advantage conferred by ALAL-1 is independent of IKBKB. On this we agree. However, as presently formulated, the sentence has "physiological" implications about the circuitry that are unwarranted. Thus, you will need to modify your conclusions to make it clear that you are not drawing conclusions about the physiological working of the circuitry but only on its impact on cancer phenotypes.

CHANGED TO "...These data suggest that, although both genes are co-regulated by NF- κ B, at least the effects observed on cancer phenotypes upon ALAL-1 depletion are not driven by IKBKB."

Next, in the discussion you state:

"Although this strategy may result in the loss of lncRNAs that act in coordination with other gene loci, it led us to identify ALAL-1, a lncRNA with bona-fide oncogenic features."

This is a bit awkward. A gene is either a bona-fide oncogene or it is not. We do not think that this manuscript is sufficient to establish ALAL-1 as a bona-fide oncogene. However it is sufficient to say that it displays oncogenic features. Thus, the word "bona-fide" should be removed.

CHANGED TO "Although this strategy may result in the loss of lncRNAs that act in coordination with other gene loci, it led us to identify ALAL-1, as a lncRNA with oncogenic features."

Be sure to include a rebuttal with your final revision which illustrates how you addressed these points.